# Functional geometry of auditory cortical resting state networks derived from intracranial electrophysiology

Matthew I. Banks[1,2]*, Bryan M. Krause[1], D. Graham Berger[1], Declan I. Campbell[1], Aaron D. Boes[3], Joel E. Bruss[3], Christopher K. Kovach[4], Hiroto Kawasaki[4], Mitchell Steinschneider[5,6], Kirill V. Nourski[4,7]

1 Department of Anesthesiology, University of Wisconsin, Madison, Wisconsin, United States of America, 2 Department of Neuroscience, University of Wisconsin, Madison, Wisconsin, United States of America, 3 Department of Neurology, The University of Iowa, Iowa City, Iowa, United States of America, 4 Department of Neurosurgery, The University of Iowa, Iowa City, Iowa, United States of America, 5 Department of Neurology, Albert Einstein College of Medicine, New York, New York, United States of America, 6 Department of Neuroscience, Albert Einstein College of Medicine, New York, New York, United States of America, 7 Iowa Neuroscience Institute, The University of Iowa, Iowa City, Iowa, United States of America

* mibanks@wisc.edu

**Data Availability Statement:** Software and data used to generate figures are freely available at https://zenodo.org/record/8235227 or DOI 10.5281/zenodo.8235227. The IRB protocol under

## Abstract

Understanding central auditory processing critically depends on defining underlying auditory cortical networks and their relationship to the rest of the brain. We addressed these questions using resting state functional connectivity derived from human intracranial electroencephalography. Mapping recording sites into a low-dimensional space where proximity represents functional similarity revealed a hierarchical organization. At a fine scale, a group of auditory cortical regions excluded several higher-order auditory areas and segregated maximally from the prefrontal cortex. On mesoscale, the proximity of limbic structures to the auditory cortex suggested a limbic stream that parallels the classically described ventral and dorsal auditory processing streams. Identities of global hubs in anterior temporal and cingulate cortex depended on frequency band, consistent with diverse roles in semantic and cognitive processing. On a macroscale, observed hemispheric asymmetries were not specific for speech and language networks. This approach can be applied to multivariate brain data with respect to development, behavior, and disorders.

## Introduction

The meso- and macroscopic organization of human neocortex has been investigated extensively using resting state (RS) functional connectivity, primarily using functional magnetic resonance imaging (fMRI) [1,2]. RS data are advantageous as they avoid the substantial confound of stimulus-driven correlations yet identify networks that overlap with those obtained using event-related data [3] and thus are relevant to cognitive and perceptual processing. RS fMRI has contributed greatly to our understanding of the organization of the human auditory cortical hierarchy [4–6], but only a few complementary studies have been conducted using

which these patients were consented specified that participants' data would be shared with qualified individuals working on approved scientific projects following establishment of a data use agreement. Intracranial EEG data are collected during a medical procedure (inpatient monitoring for seizure) and, as such, contain information that could be used to draw inferences about the medical conditions of an individual. The neural data from iEEG and fMRI, along with demographic data reported in the manuscript, have the potential for use beyond scientific inquiry were it shared without restrictions. A data transfer and use agreement between the Authors' institution and scientists with legitimate interest in the dataset reduces the risk of loss of privacy of the research participants who have contributed their neural data to this work. Thus, the complete data set is available upon request and establishment of a formal data sharing agreement. Please contact The University of Iowa Division of Sponsored Projects at dsp-contracts@uiowa.edu to request data access. Further information is available from Bryan Krause (bmkrause@wisc.edu) or the corresponding author.

**Funding:** This work was supported by the National Institutes of Health (grant numbers R01-GM109086 to MIB and to KVN and R01-DC04290 to KVN). The funders had no role in study design, data collection and analysis, decision to publish, or preparation of the manuscript.

**Competing interests:** The authors have declared that no competing interests exist.

**Abbreviations:** AGA, anterior angular gyrus; AGP, posterior angular gyrus; ATL, anterior temporal lobe; CingM, middle cingulate; CT, computed tomography; DME, diffusion map embedding; FDR, false discovery rate; fMRI, functional magnetic resonance imaging; HGAL, anterolateral Heschl's gyrus; HGPM, posteromedial Heschl's gyrus; iEEG, intracranial electroencephalography; IFG, inferior frontal gyrus; IFGop, IFG pars opercularis; IFGor, IFG orbitalis; IFGtri, IFG triangularis; InsA, anterior insula; InsP, posterior insula; ITGA, anterior portion of inferior temporal gyrus; ITGP, posterior portion of inferior temporal gyrus; MNI, Montreal Neurological Institute; MTGA, anterior portion of middle temporal gyrus; MTGM, middle portion of middle temporal gyrus; pC, precuneus; PCC, posterior cingulate; PHG, parahippocampal gyrus; PMC, premotor cortex; PP, planum polare; PreCG, precentral gyrus; ROI, region of interest; RS, resting state; SMG, supramarginal gyrus; SNR, signal-to-noise; STG, superior temporal gyrus; STGM, middle portion of the superior temporal gyrus; STGP, posterior portion of the superior

electrophysiology in humans (e.g., [7–9]). Compared to fMRI, intracranial electroencephalography (iEEG) offers superior spatiotemporal resolution, allowing for analyses that accommodate frequency-dependent features of information exchange in these networks [10,11]. For example, cortico-cortical feedforward versus feedback information exchange occurs via band-specific communication channels (gamma band and beta/alpha bands, respectively) in both the visual [11–15] and auditory [16–19] systems. There are also important regions involved in speech and language processing for which iEEG can provide superior spatial resolution and signal characteristics compared to fMRI, including in the anterior temporal lobe [20,21] and the upper versus lower banks of the superior temporal sulcus (STS) [22,23]. However, variable electrode coverage in human intracranial patients and small sample sizes are challenges to generalizing results.

We overcome these limitations using a large cohort of subjects that together have coverage over most of the cerebral cortex and leverage these data to address outstanding questions about auditory networks. We address the organization of human auditory cortex at 3 spatial scales: fine-scale organization of regions adjacent to canonical auditory cortex, clustering of cortical regions into functional processing streams, and hemispheric (a)symmetry associated with language dominance. We present a unified analytical framework applied to RS human iEEG data that embeds functional connectivity data into a Euclidean space in which proximity represents functional similarity. A similar analysis has been applied previously to RS fMRI data [24–26]. We extend this analytical approach and demonstrate methodology appropriate for hypothesis testing at each of these spatial scales.

At the fine scale, individual areas within canonical auditory cortex and beyond have different sensitivity and specificity of responses with respect to stimulus attributes [27–29]. These differences are related to underlying connectivity patterns both within the auditory cortex and with other brain areas [22]. Though there is broad agreement that posteromedial Heschl's gyrus (HGPM) represents core auditory cortex, functional relationships among HGPM and neighboring higher-order areas are still a matter of debate. For example, the anterior portion of the superior temporal gyrus (STGA) and planum polare (PP) are adjacent to auditory cortex on Heschl's gyrus yet diverge from it functionally [30,31]. The posterior insula (InsP), on the other hand, has response properties similar to HGPM yet is not considered a canonical auditory area [32]. The STS is a critical node in speech and language networks [22,33–37], yet its functional relationships with other auditory areas are difficult to distinguish with neuroimaging methods. Indeed, distinct functional roles of its upper and lower banks (STSU, STSL) have only been recently elucidated with iEEG [23].

Questions remain regarding mesoscale organization as well. The auditory hierarchy is posited to be organized along 2 processing streams (ventral "what" and dorsal "where/audiomotor") [38–40]. The specific brain regions involved and the functional relationships within each stream are vigorously debated [41–44]. Furthermore, communication between auditory cortex and hippocampus, amygdala, and anterior insula (InsA) [45]—areas involved in auditory working memory and processing of emotional aspects of auditory information [46–49]—suggests a third "limbic" auditory processing stream, complementary to the dorsal and ventral streams.

At a macroscopic scale, hemispheric lateralization is a classically described organizational feature of speech and language function [50,51]. However, previous studies have shown extensive bilateral activation during speech and language processing [52–54], and more recent models emphasize this bilateral organization [39]. Thus, the degree to which lateralization shapes the auditory hierarchy and is reflected in hemisphere-specific connectivity profiles is unknown [38,42,55–58].

temporal gyrus; STS, superior temporal sulcus; TP, temporal pole; wPLI, weighted phase lag index.

To address these questions, we applied diffusion map embedding (DME) [59,60] to functional connectivity measured between cortical regions of interest (ROIs). DME is part of a broader class of analytical approaches that leverage the spectral properties of similarity matrices to reveal the intrinsic structure of datasets [61]. When applied to multivariate neurophysiological signals, DME maps connectivity from anatomical space (i.e., the location of the recording sites in the brain) into a Euclidean embedding space that reveals a "functional geometry" [24]. In this space, the proximity of 2 ROIs reflects similarity in connectivity to the rest of the network. Implicit in the use of the term "functional" is the assumption that 2 ROIs that are similarly connected to the rest of the brain are performing similar functions. Here, we use the DME approach to provide a low-dimensional representation convenient for display while also facilitating quantitative comparisons on multiple spatial scales. We tested prespecified hypotheses of specific ROI relationships involving STSL and STSU in the gamma band using permutation tests. We applied exploratory statistical analyses to beta band connectivity, reasoning that if gamma and beta band carry feedforward and feedback information, respectively, we expect the largest differences between these 2 bands. We present key findings from all bands to explore the sensitivity of our results to the choice of specific band.

This is the first time, to our knowledge, DME analysis has been applied to electrophysiological data, which allows exploration of the band specificity of network structure. Also novel in our approach is the examination of relationships based on inter-ROI distances in embedding space, which are robust to changes in the underlying basis functions of the space.

## Results

### DME applied to iEEG data

Intracranial electrodes densely sampled cortical structures involved in auditory processing in the temporal and parietal lobes, as well as prefrontal, sensorimotor, and other ROIs in 49 participants (22 female; Fig 1, S1 and S2 Tables). A total of 6,742 recording sites (66.1% subdural, 33.9% depth) were used in the analyses. On average, each participant contributed 138 ± 54 (mean ± standard deviation) recording sites, representing 28 ± 7.7 ROIs (mean ± standard deviation) (see example in Fig 2A). Fig 1B summarizes both subdural and depth electrode coverage by plotting recording sites in Montreal Neurological Institute (MNI) coordinate space and projecting them onto an average template brain for spatial reference. Of note, assignment of recording sites to ROIs as depicted in Fig 1 was made based on the sites' locations in each participant's brain rather than based on the projection onto the template brain, thus accounting for the high individual variability in cortical anatomy (see Methods for details).

The brain parcellation scheme depicted in Fig 1A was developed based on a combination of physiological and anatomical criteria and has been useful in our previous analyses that were largely focused on auditory processing [62–67]. One goal of the analysis presented in this study is to develop instead a parcellation scheme based on functional relationships between brain areas. Accordingly, we revisit below the parcellation shown in Fig 1A with a data-driven scheme.

DME was applied to pairwise functional connectivity measured as orthogonalized power envelope correlations [68] computed between recording sites in each participant. We focus on gamma band power envelope correlations because of its established role in feedforward information exchange in the auditory system [16–19] and use gamma band as a reference in presentation of data from other bands. The functional connectivity matrix was normalized and thresholded to yield a diffusion matrix $\mathbf{P_{symm}}$ with an apparent community structure along the horizontal and vertical dimensions (Fig 2B). Within the context of DME, the term "diffusion" reflects the idea of energy or information "diffusing" through the graph being analyzed. DME

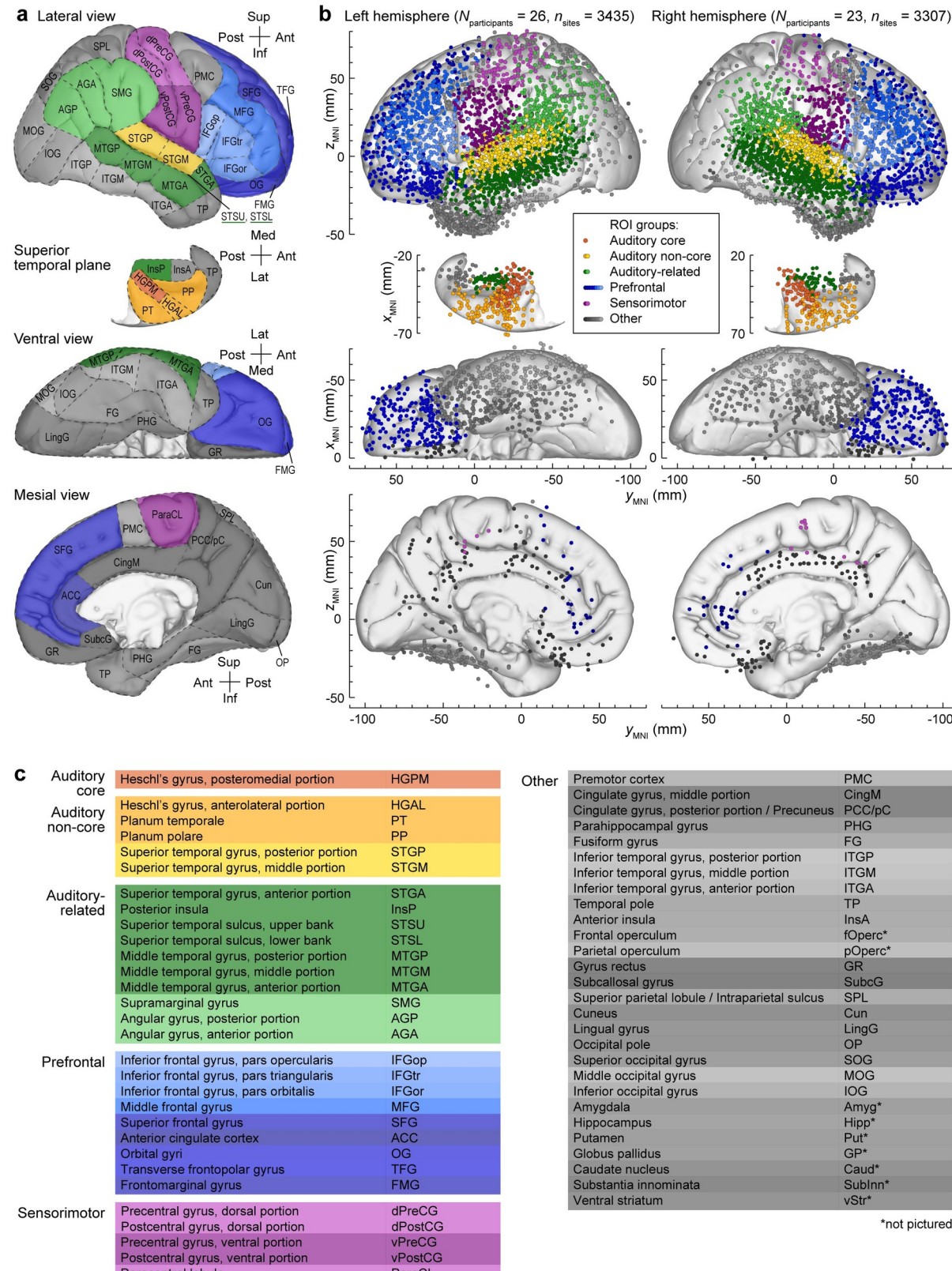

**Fig 1. ROIs and electrode coverage in all 49 participants. (a)** ROI parcellation scheme. **(b)** Locations of recording sites, determined for each participant individually and color-coded according to the ROI group, are plotted in Montreal Neurological Institute (MNI) coordinate space

and projected onto the Freesurfer average template brain for spatial reference. All depicted sites were within cortical gray matter; some appear as outside the brain due to individual variability in brain anatomy relative to the template brain. Color shades represent different ROIs within a group. Projections are shown on the lateral, top-down (superior temporal plane), ventral and mesial views (top to bottom). Recording sites over orbital, transverse frontopolar, inferior temporal gyrus, and temporal pole are shown in both the lateral and the ventral view. Sites in fusiform, lingual, parahippocampal gyrus and gyrus rectus are shown in both the ventral and medial view. See S2 Table for detailed information on electrode coverage. Sites in the frontal operculum ($n = 23$), parietal operculum ($n = 21$), amygdala ($n = 80$), hippocampus ($n = 86$), putamen ($n = 15$), globus pallidus ($n = 1$), caudate nucleus ($n = 10$), substantia innominata ($n = 5$), and ventral striatum ($n = 2$) are not visible due to the opacity of the template brain but are included in S2 Table. (**c**) ROI groups, ROIs, and abbreviations used in the present study. See S3 Table for alphabetized list of abbreviations.

reveals the functional geometry of the sampled cortical sites by using the structure of $\mathbf{P_{symm}}$ and a free parameter $t$ to map the recording sites into an embedding space. In this space, proximity between nodes represents similarity in their connectivity to the rest of the network (Fig 2C; see S1 Fig for additional views). The parameter $t$ corresponds to diffusion time: larger values of $t$ shift focus from local towards global organization. DME exhibited superior signal-to-noise (SNR) characteristics (i.e., distance between nodes versus uncertainty in node position; see Methods) compared to direct analysis of functional connectivity in 43 out of 49 participants (S2 Fig).

Functionally distinct regions are isolated along principal dimensions in embedding space. For example, in Fig 2C, auditory cortical sites (red/orange/yellow) and sites in prefrontal cortex (blue) were maximally segregated along dimension 1 (see Fig 1 and S3 Table for the list of abbreviations). Other regions (e.g., middle temporal gyrus) had a more distributed representation within the embedding space, consistent with their functional heterogeneity.

## Functional geometry of cortical networks

Electrode placement was based solely on clinical criteria in each participant and thus precise locations of recording sites varied across participants. To pool data across participants with variable electrode coverage, it was necessary to compute $\mathbf{P_{symm}}$ matrices at the ROI level and average across participants. The results for gamma band data are shown in Fig 3A. The eigenvalue spectrum $|\lambda_i|$ of this averaged $\mathbf{P_{symm}}$ showed a clear separation between the first 4 and the remaining dimensions (Fig 3A, inset), indicating that the first 4 dimensions of embedding space accounted for much of the community structure of the data. Indeed, these first 4 dimensions accounted for >80% of the diffusion distance averaged across all pairwise distances in the space, a typical measure for deciding which dimensions to retain when DME is used as a dimensionality reduction method [60]. This inflection point in the eigenvalue spectrum was identified algorithmically (see Methods) for each frequency band and yielded the number of retained dimensions $n = 6, 6, 7, 4$, and 6 for theta, alpha, beta, gamma, and high gamma bands, respectively.

The gamma band data are plotted in the first 4 dimensions of embedding space in Fig 3B, where the sizes of the ellipsoids for each ROI represent estimates of position variance across participants obtained via bootstrapping. These data provide a graphical representation of the functional geometry of all sampled brain regions (see also S3 Fig and S1 and S2 Movies; see S4 Fig for average beta band embeddings). Functionally related ROIs tended to group together, and these ROI groups segregated within embedding space. For example, auditory cortical and prefrontal ROIs were at opposite ends of dimension 1, as were visual cortical (ITGP, ITGM, LingG, FG) and prefrontal ROIs. Parietal and limbic ROIs were at opposite ends of dimension 2, and auditory and visual ROIs were maximally segregated along dimension 4. By contrast, some ROIs [e.g., STGA, anterior and middle portions of middle temporal gyrus (MTGA, MTGM), middle cingulate (CingM)] were situated in the interior of the data cloud.

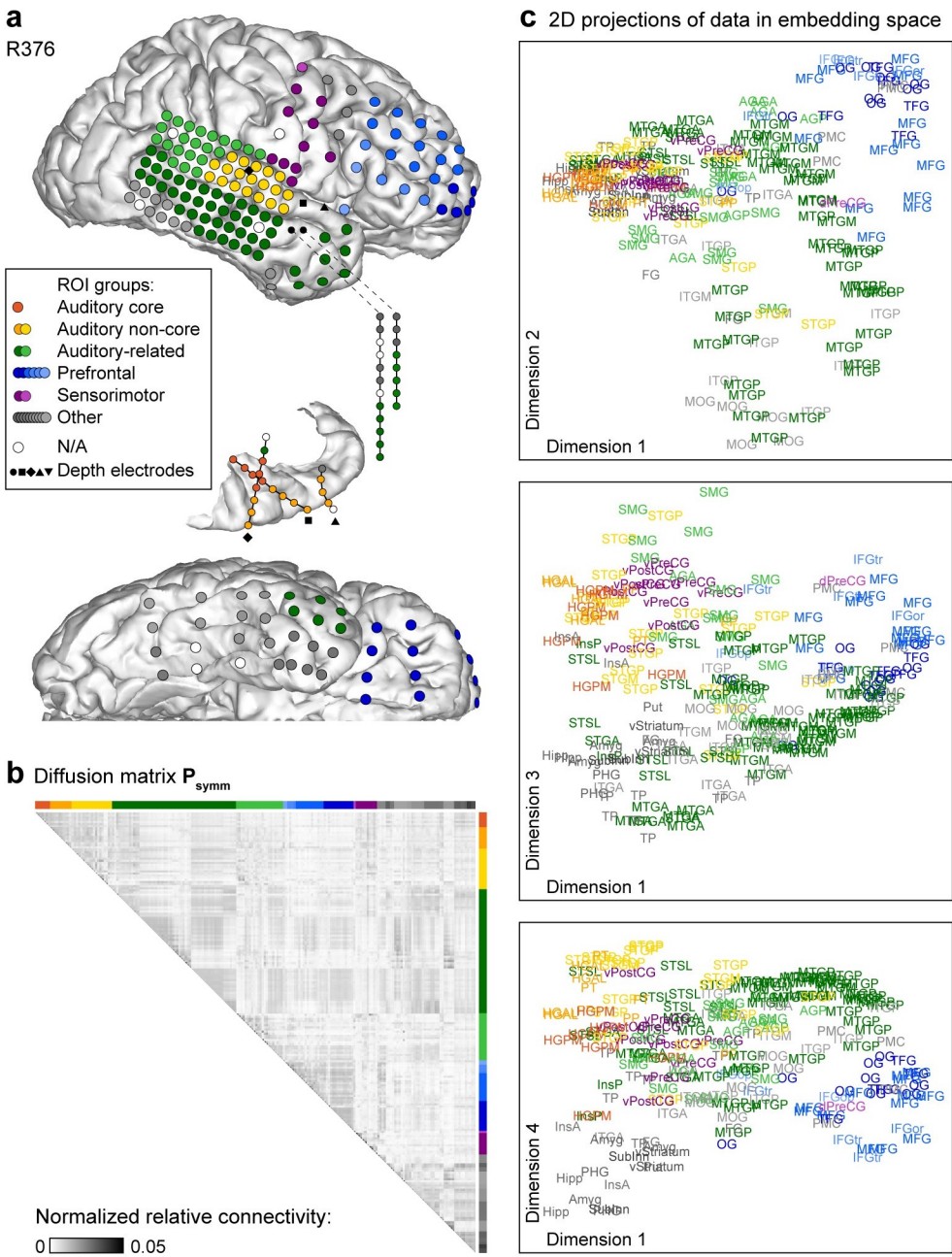

**Fig 2. Functional geometry of cortical networks revealed by DME applied to gamma band power envelope correlations in a single participant (R376).** (**a**) Electrode coverage. (**b**) Diffusion matrix $P_{symm}$. (**c**) Data plotted on the same scale in the first and second, first and third, and first and fourth dimensions of embedding space (top to bottom). Two points that are close in embedding space are similarly connected to the rest of the network and thus assumed to be functionally similar.

The data shown in Fig 4 are based on a value of the diffusion parameter $t = 1$, chosen to maximize the local network structure that can be detected in the data. To test the sensitivity of the results to this parameter choice, the analysis was repeated using the "multiscale" implementation of DME, which considers all values of $t$ simultaneously [69] (see Methods and S1 Text). The results were compared to those in Fig 4 by calculation the correlation of inter-ROI

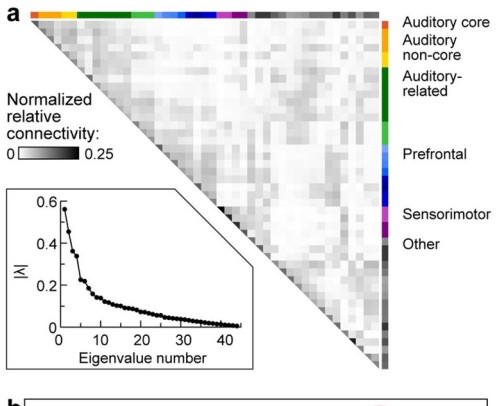

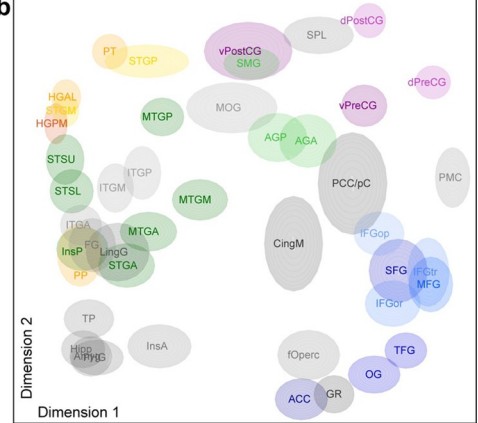

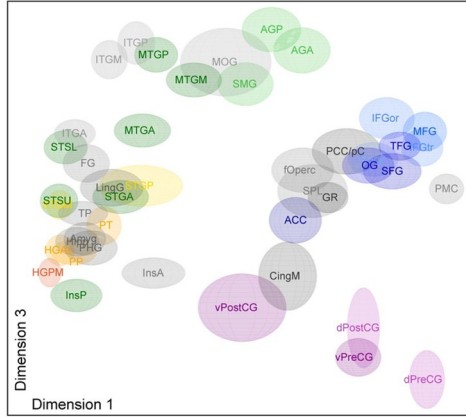

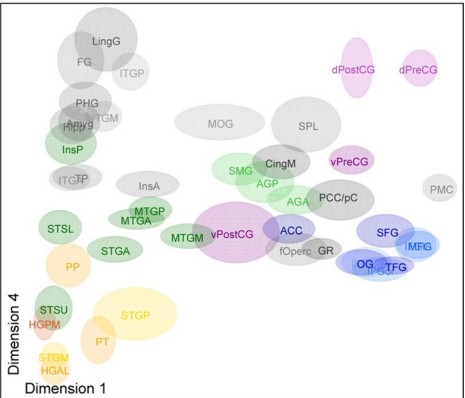

**Fig 3. Summary of functional geometry of cortical networks via DME applied to gamma band power envelope correlations.** The semiaxis lengths of each ellipsoid indicate the standard deviation of bootstrap estimates in each dimension. (**a**) Average diffusion matrix. **Inset:** Eigenvalue spectrum. (**b**) Data plotted on the same scale in the first and second, first and third, and first and fourth dimensions of embedding space (top to bottom). Estimates of variance across participants in the locations of each ROI in embedding space were obtained via bootstrapping and are represented by the size of the ellipsoid for each ROI.

distances in the 2 analyses. The correlation was extremely high (r = 0.97), indicating that the results are robust to the specific choice of the parameter *t*. Correlation values were also high for results obtained with connectivity thresholds ranging from 10% to 50% (S5A Fig), indicating that the results were also robust to the specific choice of threshold.

One advantage of applying DME to electrophysiological data is the opportunity to examine features of the embeddings that are band specific. DME applied to bands other than gamma produced similar embeddings. Inter-ROI distances were similar for adjacent bands ($r \geq 0.82$),

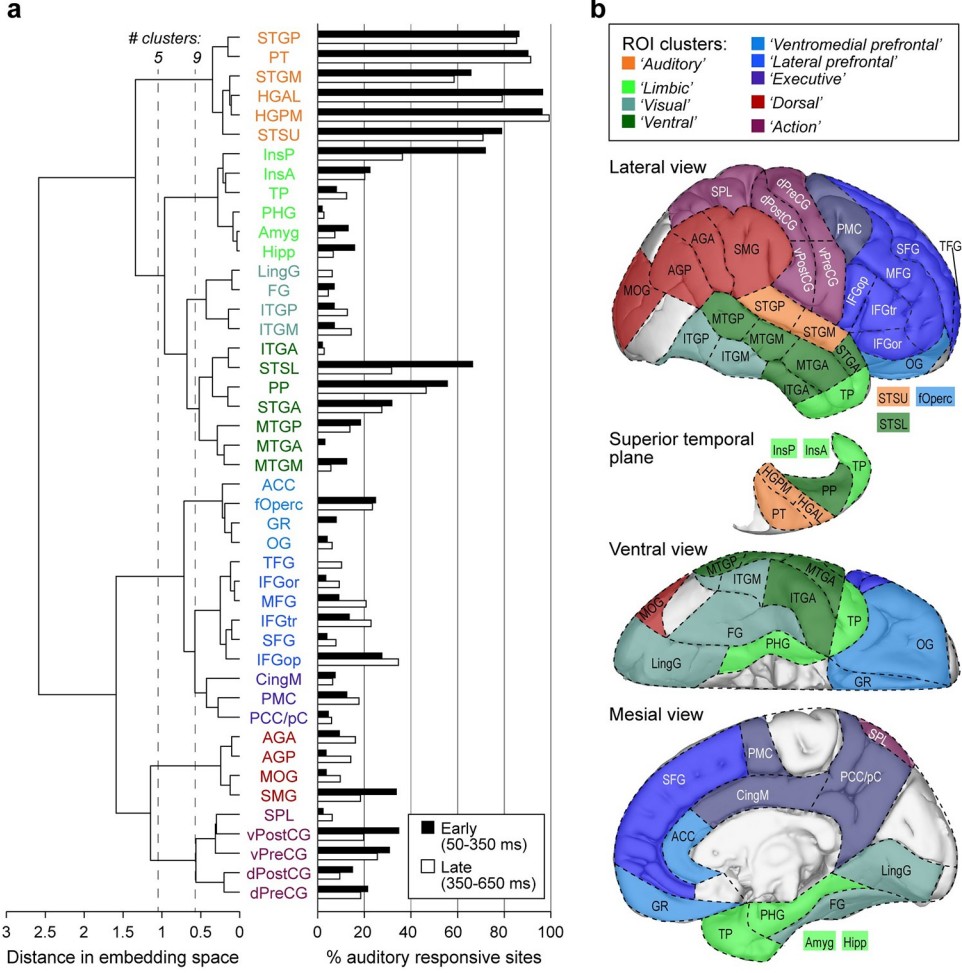

**Fig 4. Hierarchical clustering of embedding data shown in Fig 3. (a)** Left side shows linkages between ROI groups identified using agglomerative clustering. Two thresholds are denoted (vertical dashed lines), one yielding 5 clusters and one yielding 9. ROIs are colored to indicate cluster membership. Right side shows auditory responsiveness in each ROI, quantified as percentages of sites in each ROI with early (50–350 ms after stimulus onset; black bars) and late (350–650 ms; white bars) high gamma responses to 300 ms monosyllabic words. **(b)** Brain parcellation based on hierarchical clustering illustrated in (**a**).

and even for nonadjacent bands ($r \geq 0.67$; S5B Fig). Thus, DME identified some organizational features of cortical networks that were not band specific.

A second connectivity measure based on phase synchronization (debiased weighted phase lag index (wPLI) [70]) produced embeddings in the alpha and theta bands that were similar to each other (r = 0.85; S5B Fig) and to those derived from envelope correlations in those bands ($r \geq 0.73$), suggesting that common features of network structure are captured by DME. Interestingly, correlations were much lower for beta band wPLI, indicating that a phase-based connectivity measure captures distinct features in this band.

The mean correlation between inter-ROI embedding distances for original versus bootstrapped data was high in each band (r = 0.91, 0.85, 0.87, 0.88, and 0.86 for high gamma, gamma, beta, alpha, and theta, respectively). These analyses suggest that distances in embedding space are robust across participants, indicating that DME offers a robust approach to exploring functional geometry.

## DME elucidates fine-scale functional organization beyond anatomical proximity

The connectivity metric employed here discards components exactly in phase between 2 brain regions, mitigating the influence of volume conduction [68]. However, brain areas that are anatomically close to each other are often densely interconnected [71–73]. Thus, anatomical proximity is expected to contribute to the observed functional geometry. Overall, however, anatomical proximity explained only 14% of the variance in embedding distance derived from gamma band connectivity (mean adjusted $r^2 = 0.14$ for regressions between anatomical and embedding Euclidean distance, calculated separately for each ROI). Anatomically adjacent ROIs that were separated in embedding space for gamma band included STGA and STGM, temporal pole (TP) and the rest of the anterior temporal lobe (ATL), and InsA and InsP. Similar results were obtained for embeddings derived from beta band data (S4 Fig). Thus, the embedding representation elucidates organizational features beyond anatomical proximity.

## Planum polare (PP) and posterior insula (InsP) are functionally distinct from other auditory cortical ROIs

The grouping of canonical auditory ROIs is apparent in Figs 3B and S4, as PT, HGAL, and middle and posterior portions of the superior temporal gyrus (STGM, STGP) were all close to HGPM in embedding space. One notable exception, PP, located immediately anterior to anterolateral Heschl's gyrus (HGAL), segregated from the rest of auditory cortical ROIs along dimension 2 in embedding space (Fig 3B, upper panel, lower left corner; S4 Fig, upper left panel, lower left corner). This result is consistent with PP being a higher order auditory area.

In contrast, InsP is a region that is anatomically distant from HGPM yet responds robustly to acoustic stimuli [32], suggesting that a portion of this area could be considered an auditory region [74]. For example, InsP can track relatively fast (>100 Hz) temporal modulations, similar to HGPM [32,75], possibly due to direct inputs from the auditory thalamus. However, InsP was functionally segregated from HGPM and was situated between auditory and limbic ROIs, consistent with the broader role of InsP in both exteroceptive and interoceptive processing [76,77].

## Hierarchical distinction of STSU and STSL

Unlike InsP and PP, STSU was located near early auditory regions in embedding space, and for gamma band was significantly closer to auditory cortex (core and non-core ROIs; see

Fig 1) in embedding space compared to STSL (test by permutation of STSU/STSL labels, $p < 0.0001$). In beta band, the difference in distance to auditory cortex was not significant ($p = 0.051$). This distinction between STSL and STSU is consistent with differences in their response properties reported recently [23]. Particularly, responses in STSL, but not STSU, were predictive of performance in a semantic categorization task. Those results suggest that STSL would likely be closer in embedding space to regions involved in semantic processing compared to STSU. Indeed, for gamma band, STSL was significantly closer to ROIs reported to contribute to semantic processing [inferior frontal gyrus (IFG) pars opercularis/triangularis/ orbitalis (IFGop, IFGtri, IFGor), TP, STGA, MTGA, MTGP, anterior and posterior portions of inferior temporal gyrus (ITGA, ITGP), anterior and posterior angular gyrus (AGA, AGP), supramarginal gyrus (SMG)] [78–80] compared to STSU (test by permutation of STSU/STSL labels, $p = 0.0011$). Similar results were obtained in beta band ($p = 0.00044$).

## Organization of ROIs outside auditory cortex

The data of Fig 3B and S4 Fig also characterize the temporal and parietal ROIs outside auditory cortex that are nonetheless part of the extended auditory network, including components of the dorsal and ventral processing streams. These "auditory-related" ROIs (shades of green) were distributed along a considerable extent of all 4 dimensions, consistent with functional heterogeneity of these regions and their involvement in integration of sensory information from multiple modalities [81].

This heterogeneity, as well as the embedding locations of PP and STSU, suggests that DME can be used to improve the brain parcellation scheme from Fig 1. For example, MTGA in that scheme was labeled as part of the "Auditory-related" group based on its location on the lateral temporal convexity and its anatomical proximity to canonical auditory cortex. The "Other" group contains a large and diverse collection of ROIs whose relationship to auditory structures and speech and language processing is unclear. A more principled approach is warranted to arrange these and other ROIs into functional groups or streams based upon their physiology. One approach to developing such a data-driven parcellation scheme is to apply hierarchical clustering to the data in embedding space.

## Hierarchical clustering identifies mesoscale-level organizational features: ROI groups and processing streams

Hierarchical clustering was applied to the first 4 dimensions of the embedded gamma band data shown in Fig 3. The analysis illustrated a mesoscale organization of cortical ROIs (Fig 4) that aligned with the qualitative observations discussed above. As with any clustering scheme, the number of clusters is difficult to determine based on the data alone. In the left column of Fig 4A, we illustrate 2 possible thresholds yielding 5 and 9 clusters, respectively. In the 5-cluster scheme, auditory cortical ROIs (excluding PP) formed an "Auditory" cluster with STSU at one end of the dendrogram. At the other end, sensorimotor ROIs and ROIs typically considered part of the dorsal auditory stream formed clusters (labeled "Action" and "Dorsal," respectively). The remaining 2 large clusters were dominated by ventral temporal and limbic ROIs and by prefrontal and mesial ROIs (colored green and blue, respectively).

At a lower threshold, a 9-cluster scheme emerged. The ventral temporal/limbic cluster divided into 3 distinct clusters. One of these ("Limbic") included ROIs traditionally considered part of the limbic system [parahippocampal gyrus (PHG), amygdala, and hippocampus], as well as TP and the insula. A second ("Visual") included ROIs in the ventral visual stream, and a third ("Ventral") consisted of ROIs typically considered part of the ventral auditory stream. Similarly, the prefrontal cluster divided into 3 distinct clusters ("Ventromedial prefrontal,"

"Lateral prefrontal," and "Executive"). Thus, the hierarchical clustering analysis revealed a segregation of ROIs in embedding space that aligned with known functional differentiation of brain regions. Further, we can use this analysis to expand our understanding of hierarchical relationships among clusters. For example, the "Auditory" cluster was distinct from other clusters primarily in the temporal lobe but was closer to the "Limbic" cluster than "Ventral" or "Visual."

Results of hierarchical agglomerative clustering applied to data from all 5 frequency bands are shown in S6 Fig. The color scheme for the ROIs is based on the gamma band results to provide a reference for comparison across bands. Auditory cortical ROIs consistently clustered together, though the specific membership of that cluster varied slightly in alpha and beta bands. Sensorimotor ROIs consistently clustered together, usually at a considerable distance from auditory ROIs, though in high gamma band dorsal and ventral sensorimotor ROIs were separated. Prefrontal and mesial ROIs tended to cluster together in all bands, albeit at variable overall position relative to auditory and sensorimotor ROIs. PP tended to cluster with anterior temporal lobe structures, and TP with limbic structures, regardless of frequency band. Thus, the temporal scale of neuronal signaling contributes importantly to establishing the structure of functional networks, consistent with previous results [10,11,82–84].

We evaluated the sensitivity of hierarchical clustering results to the choice of specific threshold (S7A–S7E Fig). As for the analysis of results across frequency band, groups of ROIs identified as clusters with a threshold of 33% in Fig 4 tended to cluster together for other threshold values, although their relative position in the one-dimensional representation of the dendrogram tended to vary. Results obtained using the multiscale DME approach were nearly identical to those obtained with t = 1 (S7F Fig), indicating that clustering was robust to the choice of the diffusion parameter $t$ as well, consistent with results for inter-ROI distances above.

We evaluated the robustness of this clustering scheme in our dataset by calculating stability as the median normalized Fowlkes–Mallows index [85] across bootstrap iterations. The index varies between 0 (random clustering across iterations) and 1 (identical clustering across iterations). The results of the analysis as a function of threshold and frequency band indicated that stability was not strongly dependent on either band or threshold, especially for 5 or more clusters (S8A Fig). We also calculated cluster-wise stability as a function of the number of clusters for gamma band using the Jaccard coefficient [86]. Stability varied across threshold and clusters (S8B Fig). Notably, the auditory cluster was the most stable for gamma band data for both the $n_{Clust}$ = 5 and 9 results illustrated in Fig 4. By contrast, the "Executive" cluster for $n_{Clust}$ = 9 was the least stable of the group.

In addition to these RS recordings, most participants engaged in additional experiments investigating representation of acoustic stimuli in the brain [23,87–89]. We used these data to evaluate auditory responsiveness of each recording site (Fig 4A, right column) and compare these response profiles to the clustering results of Fig 4A (left column). As expected, ROIs in the auditory cluster exhibited consistently high responsiveness to auditory stimuli, while visual ROIs did not. By contrast, some clusters exhibited mixed responsiveness (e.g., InsP in the limbic cluster), possibly indicating ROIs that serve as nodes bridging auditory and other brain networks.

A brain parcellation scheme based on the gamma band clustering results is illustrated in Fig 4B. We note that as for other parcellation schemes based on functional connectivity (e.g., [2,90]), the specific threshold that is most relevant and useful depends on the questions being asked and the sample size available for hypothesis testing. For example, smaller sample sizes constrain hypothesis testing to a smaller number of brain subdivisions.

## DME identifies mesoscale topological features of cortical networks

In a network, "global hubs" integrate and regulate information flow by virtue of their centrality and strong connectivity; spokes send and receive information to/from these hubs. Identification of these nodes is critical for understanding the topology of brain networks [91], yet there is ongoing debate about effective methods for identifying hubs and spokes [92]. Here, we propose a novel approach to use DME to identify global hubs versus spokes. First, we note that the closer an ROI is to the center of the data cloud in embedding space, the more equal is its connectivity to the rest of the network. A simulated example is illustrated in Fig 5A, which depicts a network of 5 ROIs, with one serving as a global hub (Fig 5A, left panel, green). The network structure can also be represented as an adjacency matrix, wherein the hub ROI has strong connectivity with other ROIs (Fig 5A, middle panel). In embedding space, this ROI occupies a central location, with the other 4 serving as spokes, i.e., nodes that interact with each other through the central hub (Fig 5A, right panel).

However, a node's proximity to the center of the data cloud reflects the homogeneity of its connectivity to the rest of the network, not necessarily the strength of that connectivity. In theory, a node could appear at a central location if it is weakly but consistently connected to all other nodes. To determine whether this occurs in our dataset, we computed the Euclidean distance from the center of embedding space and mean connectivity for all of the ROIs in Fig 3B. We show in Fig 5B a strong inverse relationship between these 2 measures. ROIs close to the center of embedding space also exhibited strong mean connectivity, suggesting that global hubs can be identified in these data using distance from the center of embedding space alone.

The identity of global hubs, and the extent to which specific nodes act as global hubs, varied across frequency bands. In the high gamma and gamma bands, ROIs presenting most strongly as global hubs included MTGA, STGA, and MTGM. ITGA, CingM, posterior cingulate/precuneus (PCC/pC), PP, fOperc, and STSL also exhibited hub-like properties. By contrast, the ROIs that were farthest from the center of embedding space were mostly unimodal sensory and motor regions, consistent with their roles as spokes in the network. The positioning of these ROIs in embedding space and their roles as spokes are also indicated by the position of these ROIs at the edges of the 1D representation depicted in the dendrograms of Figs 4 and S6.

In lower frequency bands, by contrast, CingM along with MTGM, InsP, and InsA, presented most strongly as global hubs, with the addition of ACC in the theta band. These results are consistent with network organization depending on temporal scale and are consistent with previous reports showing that mesial cortical structures regulate information flow on slower time scales [10]. Thus, DME can identify band-specific topological features critical to information flow within cortical networks.

## Differences between language-dominant and nondominant hemispheres are not specific to auditory-responsive and language-specialized ROIs

On a macroscopic scale, speech and language networks are lateralized in the human brain, with nearly all right-handed and most left-handed individuals left hemisphere language-dominant [93]. However, both hemispheres are activated during speech processing [33,39,56,94], and the extent to which lateralization is reflected in asymmetries in the organization of RS auditory networks is unclear. We hypothesized that that we would observe asymmetry in gamma band data, specifically that ROIs would be in different positions in embedding space in the language-dominant versus nondominant hemisphere. We investigated this issue by comparing the functional geometry of cortical networks derived from participants with electrode coverage in the language-dominant ($N = 24$) versus nondominant ($N = 22$) hemisphere. ROIs in the 2 hemispheres exhibited a similar functional organization in embedding space (S9 Fig).

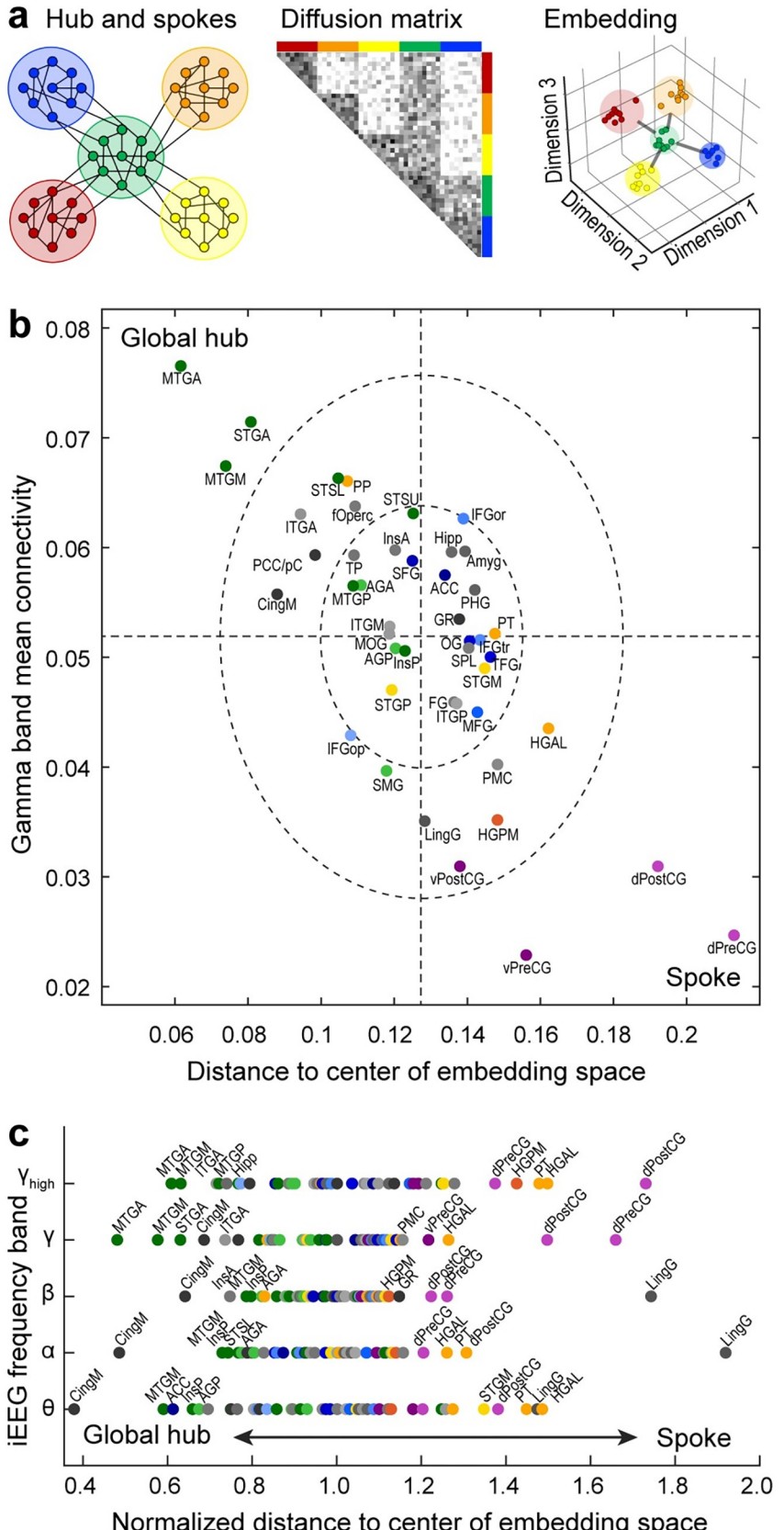

**Fig 5. Identification of network hubs.** (**a**) Schematic example illustrating the central positioning of global hubs in embedding space. (**b**) ROIs from average embedding are plotted according to their mean connectivity to the rest of the network versus their Euclidean distance to the centroid of the data cloud in the first 4 dimensions of embedding space. Dashed lines denote across-ROI means. Dashed ellipses represent 1 and 2 standard deviations from the mean. (**c**) Distance to center of embedding space for each ROI in the 5 studied frequency bands. Distances are normalized to the median distance within each band to allow for comparison across bands.

Permutation analysis indicated that for gamma band, the positions of ROIs in embedding space were not significantly different between dominant and nondominant hemispheres (all *p*-values > 0.05). Furthermore, there was no significant correlation between the change in position in embedding space and either early or late auditory responsiveness (early: *p* = 0.94; late: *p* = 0.86; Fig 6A). Similar results were obtained in exploratory analyses of beta band data, though 1 ROI (MTGP, *p* = 0.013) did survive false discovery rate (FDR) correction for difference in position between the 2 hemispheres.

We also analyzed inter-ROI distances to determine whether functional interactions between ROIs were different in the 2 hemispheres. For gamma band, pairwise inter-ROI distances in embedding space, calculated separately for dominant versus nondominant hemisphere, were highly correlated (*r* = 0.88), with no obvious outliers (Fig 6B, left panel). The data shown in Fig 6A have a slope <1, indicating that inter-ROI distances are consistently longer in the dominant hemisphere compared to the nondominant hemisphere (*p* = 0.0032). This multiplicative scaling of the distances is consistent with the data occupying a larger volume in embedding space for the dominant versus nondominant hemisphere, suggesting a greater functional heterogeneity for the language-dominant side of the brain. After accounting for this multiplicative scaling effect, following FDR correction, there were no specific inter-ROI distances that were significantly different between the 2 hemispheres. Similar results were obtained in exploratory analyses of beta band data (pairwise inter-ROI distances *r* = 0.79; longer inter-ROI distances in dominant hemisphere *p* = 0.0071; no pairwise distances significant after FDR correction).

When considering ROIs specifically involved in speech and language comprehension and production [PT, PP, STSL, STGP, STGM, STGA, SMG, AGA, premotor cortex (PMC), precentral gyrus (PreCG), IFGop, IFGtr] [36,42,95], the correlation in pairwise inter-ROI distances in embedding space was also high (*r* = 0.90; Fig 6B). Furthermore, the data in Fig 6B exhibited a similar multiplicative scaling as observed for all the ROIs shown in Fig 6A. Indeed, the slope for the data in Fig 6B was indistinguishable from the slope for the data in Fig 6A (*p* = 0.93). Similar results were obtained in exploratory analyses of beta band data (pairwise inter-ROI distance correlations, *r* = 0.76; difference between speech and language ROIs versus others, *p* = 0.35). Thus, hemispheric asymmetry of functional organization specific to speech and language networks was not detectable in RS connectivity.

## Comparison to embeddings derived from RS-fMRI data

So far, we have presented results at multiple spatial scales based on intracranial electrophysiology. However, these intracranial recordings sample the brain nonuniformly and sparsely as dictated by clinical considerations. This feature presents problems at 2 spatial scales: First, cortical regions are not sampled uniformly (with some not sampled at all). Second, ROIs are not sampled uniformly across their volume. To examine the impact of these sampling issues, we compared iEEG-based DME to DME applied to RS-fMRI data available in a subset of 10 participants (1,591 recording sites).

We first tested the consistency of functional geometry derived from the 2 modalities in the same participants (Fig 7). Connectivity matrices were constructed based on RS-fMRI data

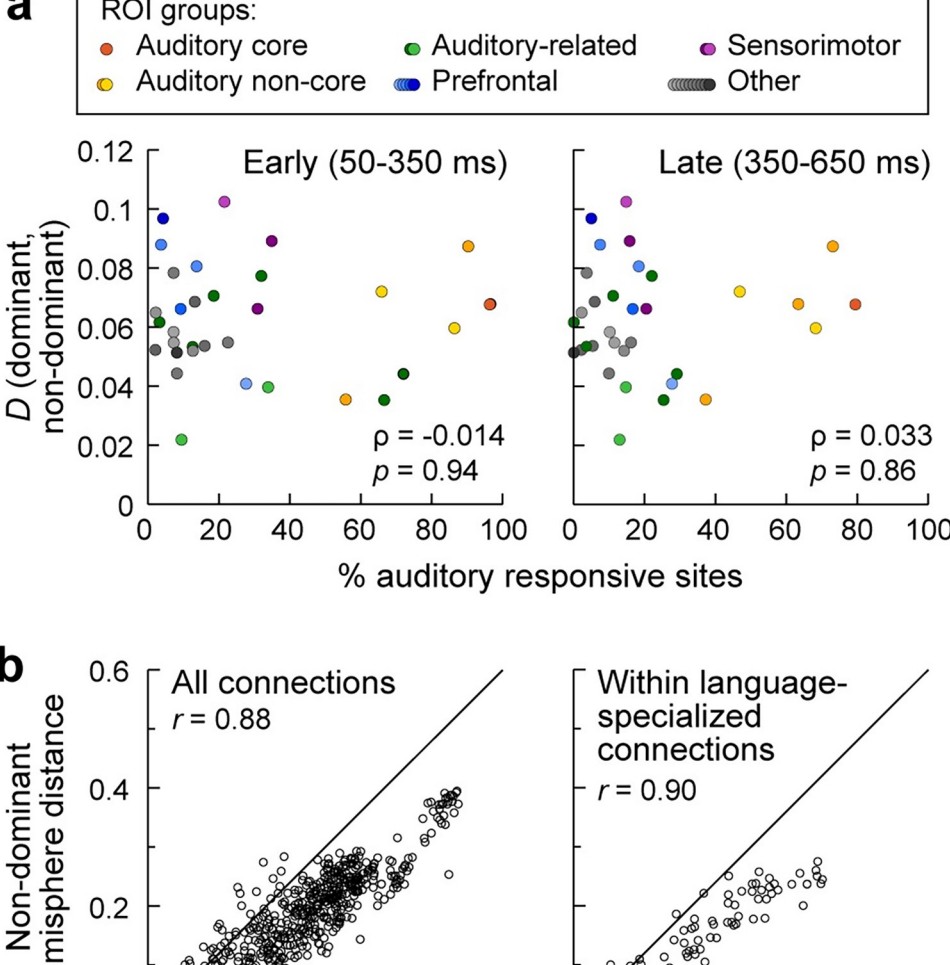

**Fig 6. Hemispheric asymmetries in RS connectivity are not driven by auditory-responsive and language-specialized ROIs.** Inter-ROI distances in embedding space for nondominant versus dominant hemisphere participants. (**a**) Comparison between the change in position in embedding space from dominant to nondominant hemisphere and the auditory responsiveness of individual ROIs. Two-tailed Spearman's rank tests did not reveal a significant correlation between ROI asymmetry and percentage of either early or late auditory responsive sites within the ROI (left and right panel, respectively). (**b**) Pairwise distances between all ROIs and between ROIs involved in speech and language perception and production (PT, PP, STSL, STGP, STGM, STGA, SMG, AGA, PMC, PreCG, IFGop, IFGtr) are shown in the left and right panel, respectively. Note that after splitting the data into the 2 subsets (dominant and nondominant) STSU did not meet the inclusion criteria for analysis presented in the right panel (see Methods, S2 Table).

from voxels located at iEEG recording sites and grouped into the same ROIs as in Fig 1. The fMRI and iEEG embeddings averaged across participants were qualitatively similar (Fig 7A and 77B, respectively), and the overall organization derived from this subset was consistent with that observed in the full iEEG dataset (cf. Fig 3B). Inter-ROI distances in the fMRI and iEEG embedding spaces were correlated (Fig 7C). These correlations varied across band, with highest correlations for gamma and high gamma band envelopes ($r > 0.45$; Fig 7D, line and symbols), consistent with previous reports [82,96].

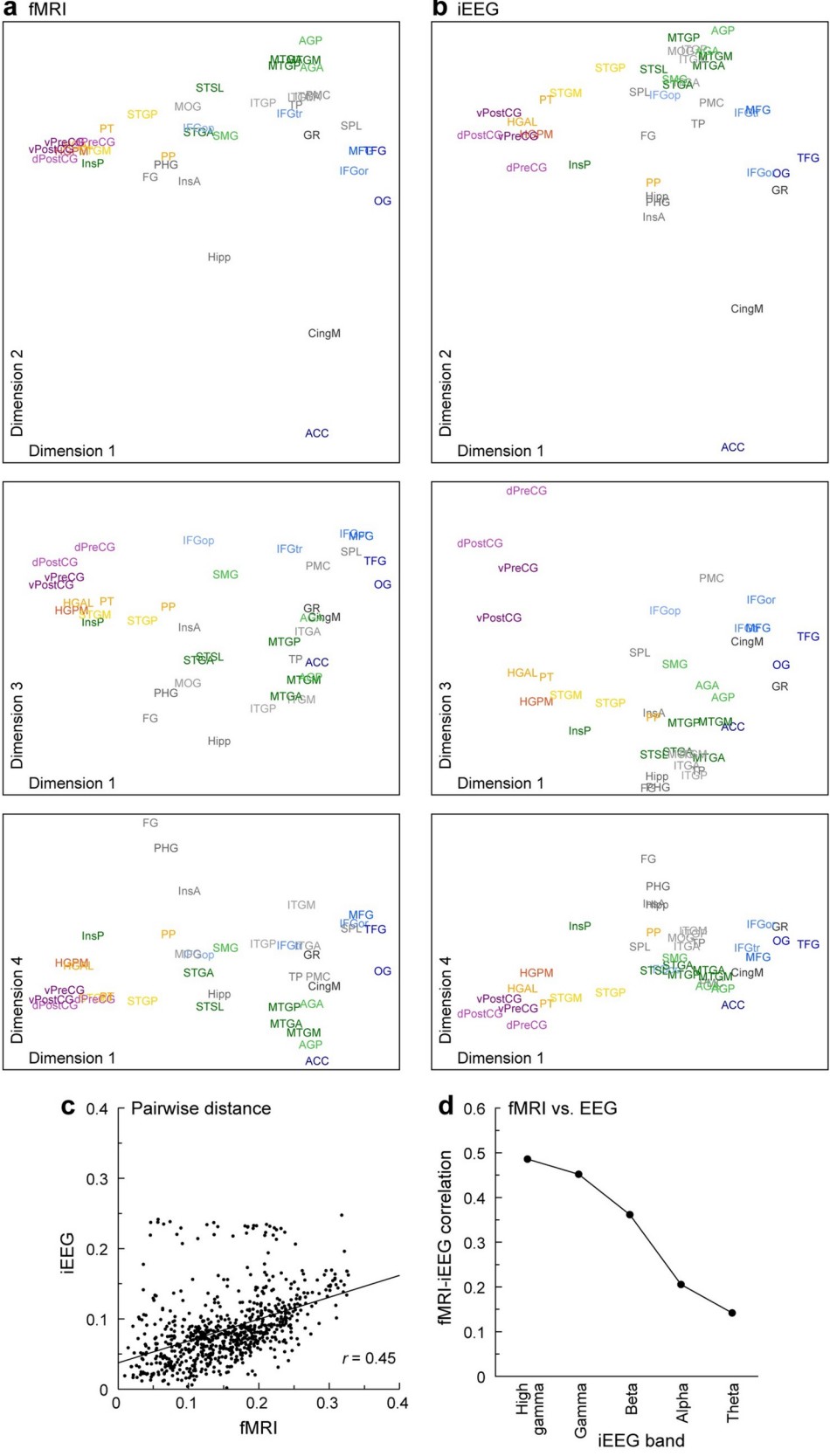

**Fig 7. Comparison of iEEG and fMRI connectivity data in embedding space.** (**a**) Participant-averaged embeddings for fMRI. (**b**) Participant-averaged embeddings for iEEG (gamma band power envelope correlations). (**c**) Inter-ROI embedding distances computed from the data in (**a**) and (**b**). (**d**) Summary of distance correlations at each frequency band.

The analysis presented in Fig 7 provide a context for using fMRI data to address questions regarding the effects of limited, nonuniform sampling. We used a standard parcellation scheme developed for fMRI data (Schaefer–Yeo 400 ROIs; [90]) rather than the iEEG parcellation scheme introduced in Fig 1, as the latter scheme does not cover the entire brain and cannot be applied to automatically parcellate the fMRI data on a voxel-by-voxel basis.

The first question we addressed was the effect of nonuniformly sampling only a subset of brain regions. For each participant, embeddings were derived from RS-fMRI connectivity matrices computed from all cortical ROIs (Fig 8A, "Full fMRI," first column). From these embeddings, we selected only points in embedding space corresponding to ROIs sampled with iEEG (Fig 8A, "Full fMRI (iEEG subset)," second column). We also computed embeddings for each subject from only the fMRI ROIs sampled with iEEG in that subject ["Partial fMRI (ROI level)," Fig 8A, third column]. We compared these embeddings to the "Full fMRI (iEEG subset)" embeddings by computing the correlation between inter-ROI distances (Fig 8B). Although the scale of the embeddings was different for the full fMRI versus partial fMRI data (because the number of dimensions was different), the two were highly correlated (median $r = 0.90$; Fig 8C). Thus, embeddings constructed from the portion of the brain sampled by iEEG were quite similar to embeddings derived from the whole brain.

The second question we addressed was the effect of representing an entire ROI by sparse sampling with a limited number of electrodes. We computed embeddings from the voxel averages across entire ROIs in each participant ["Partial fMRI (ROI level)," Fig 8A, third column] and from averages of the voxels in grey matter spheres around iEEG recording sites ["Partial fMRI (site level)," Fig 8A, rightmost column]. ROI- and site-level embedding distances were strongly correlated (median $r = 0.65$; Fig 8C).

Thus, sparse sampling within an ROI had a greater impact on estimates of functional geometry than limited sampling of the complete set of ROIs. Overall, however, ROIs were faithfully represented in embedding space even when DME was based on a small number of locations within ROIs. Taken together, these results indicate broad consistency between functional organization derived from iEEG and fMRI and the robustness of this approach to sparse sampling afforded by iEEG recordings.

## Discussion

### Organization of auditory cortical networks

We have shown that DME applied to iEEG data can be used to characterize the organization of the human auditory cortical hierarchy at multiple spatial scales. We demonstrate methodology for testing specific hypotheses about gamma band data at each of these scales using DME. We also use exploratory analyses (e.g., hierarchical clustering, analyses of other frequency bands) to generate data-driven hypotheses for study using future data sets.

### Investigating cortical network organization using resting state data

The results presented here are based on the analysis of RS (i.e., task-free) data. Relationships between brain signals recorded at different locations derive from synaptic connections between neurons in those locations. Thus, these data provide valuable information about the underlying brain organization despite the absence of a task or a controlled sensory stimulus.

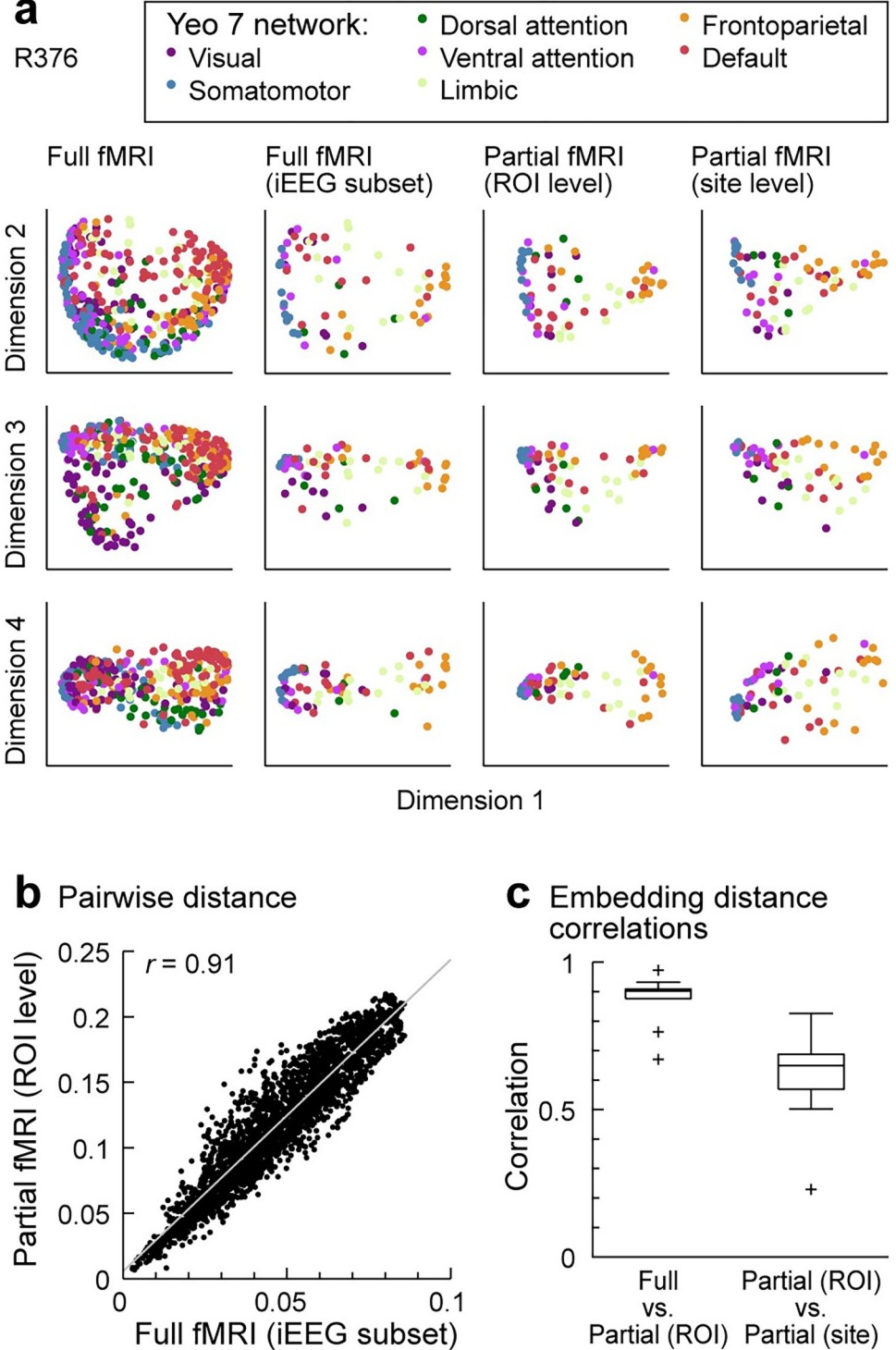

**Fig 8. Comparison of embeddings derived from full fMRI connectivity matrices and connectivity matrices computed using only ROIs sampled with iEEG.** (**a**) Data in the first 4 dimensions of embedding space for a single participant. Shown are embeddings of all derived from the full RS-fMRI connectivity matrix (first column); the subset of the data points in the first column corresponding to ROIs sampled via iEEG (second column); and embeddings derived from connectivity matrices including only the ROIs sampled via iEEG, calculated by averaging across the entire ROI (third column), and calculated based on the specific recording sites in that participant (fourth column). (**b**) Comparison of embedding distances calculated from the full fMRI embedding (i.e., data in (**a**), second column) versus distances calculated from the partial fMRI embedding (i.e., data in (**a**), third column). (**c**) Summary across participants

of distance correlations between full fMRI embeddings versus partial embeddings calculated based on the entire ROI (left: "Full vs. Partial (ROI)") and between partial embeddings calculated based on the entire ROI versus those calculated based on recording sites [*i* "Partial (ROI) vs. Partial (site)"].

The same areas that are coactivated during sensory processing exhibit RS connectivity with each other, and RS networks map onto relevant behavioral and task-related domains [2,3]. Numerous previous studies based on BOLD fMRI have used analyses of RS activity to gain insight into the organization of human brain networks and how this organization is altered due to brain disorders, during development and ageing, and in response to pharmacological treatments [22,26,97–101].

Connectivity can also be measured in the context of event-related paradigms, including behavioral tasks [3,102]. These analyses have provided important insights into the relationship between network structure and function [103], especially in the context of attention [104]. A key advantage of RS analyses is that they are based on data that are far more stationary compared to data derived from task-based experiments. In the case of sensory regions, this avoids a confound inherent to investigations of connectivity in the presence of a stimulus, which itself would produce correlated activity at directly driven sites, i.e., 2 unconnected sites in the brain driven by the same sensory stimulus will exhibit apparent connectivity solely due to the stimulus, in the absence of a physical connection between the sites. Additionally, RS analyses typically can draw on considerably more data than is available from task-based experiments, allowing for better estimates of connectivity.

As in previous studies, we provide a snapshot of the organization of these networks, corresponding to a static representation. However, these networks are dynamic due to short- and long-term plasticity driven both by internal (e.g., changes in arousal state) and external factors (e.g., sensory stimulation and directed behavior). The organization derived from studies such as this one provides a framework for understanding these dynamics.

## Frequency band-specific properties of cortical networks

Previous reports have shown that cortical networks defined by functional or effective connectivity derived from electrophysiological data exhibit organizational structure that depends on the frequency band being analyzed. This manifests in 2 ways relevant to the results presented here. First, canonical RS networks originally identified using RS BOLD fMRI data vary across band in the strength of within-network connectivity and in the relationship between electrophysiological- and fMRI-derived connectivity networks [10,82]. Second, detailed analyses of the relationship between anatomical projection patterns and functional or effective connectivity indicate that especially in auditory and visual cortical hierarchies, feedforward information streams rely on connectivity primarily in gamma and high-gamma bands, while lower frequency bands (alpha, beta) underlie feedback connectivity [11–19]. Based on these previous reports, we suggest that the gamma band organization that is the focus of the current report reflects feedforward connectivity. Indeed, the auditory responsiveness profile depicted in Fig 4B is most strongly predicted by clustering analysis applied to gamma and high-gamma band data in embedding space. Results for other bands differed from gamma band results especially in the identity of network hubs (Fig 5), where CingM cortex emerged as the ROI with the most pronounced "hubness." By contrast, the overall organizational features considered at various spatial scales did not differ strongly between bands, suggesting that while temporal scale is an important contributor to network organization, functional connectivity on these different scales tends to overlap. In the case of comparisons between feedforward and feedback networks, this is consistent with the tendency of cortical areas to be coupled bidirectionally [105].

### Fine scale: Organization of auditory cortex

At a fine spatial scale, previous work in nonhuman primates has defined over a dozen auditory cortical fields based on cytoarchitectonics, connectivity, and response properties [106]. By contrast, there is no consensus on how auditory cortex is organized in humans, with multiple parcellations based on cytoarchitectonics, tonotopy, or myeloarchitecture [107–110]. Our results contribute to this body of knowledge by showing that several superior temporal ROIs including core auditory cortex (HGPM) and putative auditory belt and parabelt areas (PT, HGAL, STGM) [107,110] group together in embedding space across all frequency bands. Thus, in spite of their diversity in processing of specific features of acoustic signals, these ROIs are positioned at a similar level in the auditory cortical hierarchy. Other regions, such as STGP and STSU, group with these cortical ROIs in theta, gamma, and high-gamma, but not in alpha and beta. For gamma band, proximity of STGP and STGM to HGPM in embedding space is consistent with previous studies that interpret these regions as relatively early non-core auditory cortex [29,111,112]. By contrast, although PP is anatomically close and connected to HGPM [113], for both gamma and beta band, it was not close to HGPM in embedding space. PP is distinguished among auditory cortical regions for its syntactic-level language processing [30] and its preferential activation by music, which has a strong affective component [31]. This functional differentiation is reflected in its segregation from the group of auditory cortical ROIs in embedding space.

### Fine scale: Functional differentiation between STSU and STSL

The STS is a critical node in speech and language networks linking canonical auditory cortex with higher-order temporal, parietal, and frontal areas [22,33–37]. Previous studies have shown that STSU and STSL differ in cytoarchitecture [114] and have distinct responses to speech [27,57,115,116]. A recent iEEG study demonstrated enhanced, shorter latency responses in speech syllables in STSU compared to STSL [23]. STSU is traditionally not considered part of canonical auditory cortex (but see [108]), yet it was located close to auditory cortical ROIs in embedding space in gamma band. STSL, by contrast, was closer in embedding space to semantic ROIs in both beta and gamma bands. This is consistent with iEEG evidence that responses in STSL, but not STSU, correlated with performance on a semantic categorization task [23]. The regions specifically involved in semantic processing is a current topic of debate, with multiple competing models [21,78–80]. We defined a list of semantic processing regions by combining across these models. Taken together, the results firmly place STSU and STSL at different levels of the auditory cortical hierarchy defined by gamma band connectivity.

### Mesoscale: Functional and theoretical framework of a limbic auditory pathway

Multiple lines of evidence support a pathway linking auditory cortical and limbic structures [117–120] that subserves auditory memory [45,48,49] and affective sound processing [121]. The data presented here contribute to our understanding of this pathway. Clustering analysis identified a set of ROIs including structures classically labeled as limbic (PHG, Amy, Hipp) as well as insula (InsP, InsA) and TP positioned close to the auditory cluster in embedding space for both gamma and beta bands (Figs 4 and S4). This suggests a close functional relationship that could form the basis for a limbic stream. InsP, with strong auditory responsiveness and overlapping response properties with HGPM, is likely involved in the transformation of auditory information in auditory cortex to affective representations in InsA [32]. Thus, InsP could serve as critical linking node between auditory and limbic structures.

TP is involved in semantic processing [21,30] and auditory memory [122], in particular the representation and retrieval of memories for people, social language, and behaviors ("social knowledge") [123]. Tight clustering of TP with limbic ROIs in embedding space is consistent with its previously reported functional association with limbic cortex [124,125], with which TP shares key features of laminar cytoarchitecture and strong connectivity [126]. We suggest that the organization depicted in Figs 3 and 4, combined with evidence for bidirectional information sharing between auditory cortex and limbic areas, merits the identification of a third auditory processing stream alongside the dorsal and ventral streams [38,127]. This "limbic stream" would underlie auditory contributions to affective and episodic memory processing.

## Mesoscale: Ventral and dorsal streams linking auditory and frontal cortex

Current models of speech and language processing posit the existence of ventral and dorsal processing streams linking non-core auditory cortex with PMC and IFG via several distinct anatomical pathways encompassing temporal, parietal, and frontal cortex [36,38–40]. Despite substantial experimental evidence supporting these models, there is a lack of consensus on the specific functions subserved by the 2 streams. For example, while there is consensus that the ventral stream subserves auditory object identification ("what" processing), the dorsal stream has been envisioned to subserve spatial processing ("where"; [38]) and audiomotor processing [39]. There is a parallel debate about the specific cortical regions comprising the 2 streams.

As broadly predicted by these models, temporal and parietal ROIs segregated in embedding space in the analysis presented here (Figs 3, 4, S4, and S6). Across frequency bands, we observed a cluster that included STSL and ATL ROIs, in conformity with the ventral auditory stream proposed by Hickok and Poeppel [39] and Friederici [40]. By contrast, the cluster that included SMG, AGP, and AGA aligned with the dorsal processing stream as proposed by Rauschecker and Scott [38]. The proximity of these dorsal ROIs to sensorimotor ROIs is consistent with sensorimotor contributions to dorsal stream processing [43,128]. Association of FG and MOG with the ventral and dorsal clusters, respectively, likely represents the sharing of information across sensory modalities. For example, visual information has been shown to contribute to processing in the ventral ("what") pathway [129,130].

A previous fMRI-based DME study found that primary sensory and default mode ROIs segregated along the first dimension in embedding space [25]. Coverage of mesial cortex in our dataset was limited, precluding a direct comparison. However, the striking separation between auditory and prefrontal cortex in embedding space shown here indicate that the current results align well with the previous report. This separation places auditory and frontal regions at opposite ends of a cortical hierarchy, linked by ventral and dorsal processing streams [38–40].

## Mesoscale: Network hubs

Hubs in brain networks play a critical role in integrating distributed neural activity [91,131]. In the present analysis, global hubs were characterized by their central location within embedding space (Fig 5). In the gamma band, these hubs included STGA and MTGA, both components of the ATL. Previous reports indicate that ATL serves as a network hub, transforming sensory domain-specific to domain-general representations [21,132,133] and playing a central role in semantic processing and social memory [21,123,134]. MTGM also appears as a global hub, even though it is anatomically distinct from the ATL. Interestingly, patients with semantic dementia have ATL degeneration [135,136], but the damage is often more widespread and can include MTGM [137].

Cingulate cortical ROIs (CingM, ACC) and insula were identified as hubs in lower frequency bands. CingM and ACC are active during a wide array of emotional and cognitive

processes [138,139], both consistent with their previous characterization as network hubs [131]. The identification of hubs specific to each frequency band supports the model in which the temporal scale of communication in the brain supports distinct functional networks [82–84,140]. Also consistent with this model is the frequency band-specific correspondence between iEEG and fMRI connectivity observed here (Fig 7D) and in previous reports [82,96]. Strong correspondence between BOLD fMRI connectivity and higher frequency band envelope correlations in iEEG are observed, while the correspondence for theta and alpha bands is usually still positive but lower in magnitude. Of note, this frequency dependence of connectivity is distinct from previous observations of a frequency-dependent correspondence between iEEG power and BOLD fMRI signal magnitude [141]. This relationship for connectivity also depends on brain location [82]. Because we did analyze the relationship between fMRI and iEEG in a region-specific manner, the results presented here represent an average analysis over all sampled brain regions.

Unlike other ATL structures, TP does not present as a global hub in any frequency band (Fig 5C). The close association of TP with limbic structures in embedding space suggests that TP mediates interactions between integration centers in the ATL and structures subserving memory functions. More broadly, the heterogeneity of ATL ROIs in terms of their global hub-like connectivity profiles conforms to the observation that the terminal fields of white matter tracts converging in the ATL only partially overlap [21,142,143].

## Macroscale: Hemispheric lateralization

Although speech and language networks are classically described as highly lateralized, imaging studies have demonstrated widespread bilateral activation during speech and language tasks [52–54]. Indeed, a recent fMRI study showed RS connectivity patterns in lateral temporal cortex that were comparable between left and right hemispheres [6]. We found evidence for hemispheric differences in RS cortical functional organization based on analysis of all sampled brain regions, with inter-ROI distances being systematically greater in embedding space for the language-dominant hemisphere (Fig 6B). This is consistent with greater interregional heterogeneity in that hemisphere compared to the nondominant side. Importantly, the observed asymmetry could not be attributed specifically to ROIs involved in speech and language processing (Fig 6B), nor was the difference in position in embedding space related to auditory responsiveness (Fig 6A).

Recent studies that identified hemispheric differences in RS connectivity for the STS [22] and semantic networks more broadly [144] may reflect the general asymmetry observed here. This asymmetry may relate as well to the dichotomy between domain-specific (e.g., sensory processing) and domain-general (e.g., attention, working memory) cortical systems. In particular, studies have emphasized that domain-general systems also exhibit hemispheric laterality [145,146], suggesting that the asymmetry observed here may reflect this broader organization feature. This does not exclude the possibility of asymmetries specific to auditory regions emerging during sensory tasks, for example, reflecting hemispheric biases in spectral and temporal processing [39,42].

## Caveats and limitations

A key concern regarding all human iEEG studies is that participants may not be representative of a healthy population. In the present study, results were consistent across participants despite differences in seizure disorder histories, medications, and seizure foci and aligned with results obtained previously in healthy participants [25]. Another caveat is that our dataset, however extensive, did not sample the entire brain, and it was not possible to infer connectivity with

unsampled regions. To address this, we applied DME analysis to fMRI data to establish that the organization of ROIs in embedding space was robust to the exclusion of unsampled ROIs. Although there was a greater effect of sparse, nonuniform sampling within an ROI, there was still considerable similarity in functional organization to embeddings derived from averages across the entire ROI.

While subcortical structures (e.g., thalamus) that link sensory and higher-order networks [147] were not sampled, the functional organization presented here was likely influenced indirectly by thalamo-cortical pathways [29,148]. Previous fMRI studies of RS networks focused exclusively on cortical ROIs and did not consider the role of the thalamus and other subcortical structures. Despite this limitation, these studies have yielded valuable insights into the functional organization of the human cortical networks [1,149].

Finally, we note that while most of the analyses presented here were data driven, the initial step of assignment of recording sites to specific ROIs was not. Unlike noninvasive neuroimaging methods such as fMRI, intracranial electrode recording sites sample activity discontinuously and at discrete locations. In addition, in order to pool data across participants, we were constrained by variable electrode coverage across participants. Thus, we were unable to consider structure at the finest scales due to these limits in sampling. Instead, we investigated higher levels of organization of brain regions. Our results provide insights into the relationship between network structure and function that can inform future studies with more uniform sampling.

## Concluding remarks and future directions

This study extends the DME approach to characterize functional relationships between cortical regions investigated using iEEG recordings. These data help resolve several outstanding issues regarding the functional organization of human auditory cortical networks and stress the importance of a limbic pathway complementary to the dorsal and ventral streams. These results lay the foundation for future work investigating network organization during active speech and language processing. The superior time resolution of electrophysiological data allows for dynamic connectivity analysis on time scales relevant to this processing. An important next step for this work is to adapt this analysis to scalp EEG recordings, which offer considerable advantages over fMRI in terms of accessibility and cost. While the current work focused on auditory cortical networks, this approach can be readily generalized to advance our understanding of changes in brain organization during sleep and anesthesia, disorders of consciousness, as well as reorganization of cortical functional geometry secondary to lesions.

## Materials and methods

### Ethics statement

Research protocols aligned with best practices recently aggregated in [150] and were approved by the University of Iowa Institutional Review Board (protocols #201911084 "Research of Physiology of Human Brain" and #200112047 "Human Brain Physiology Research") and the National Institutes of Health (grant #R01-DC04290); written informed consent was obtained from all participants. Research participation did not interfere with acquisition of clinically necessary data, and participants could rescind consent for research without interrupting their clinical management.

### Participants

The study was carried out in 49 neurosurgical patients (22 females) diagnosed with medically refractory epilepsy. The patients were undergoing chronic invasive electrophysiological

monitoring to identify seizure foci prior to resection surgery (S1 Table). All participants except two were native English speakers. The participants were predominantly right-handed (42 out of 49); 6 participants were left-handed, and one had bilateral handedness. The majority of participants (35 out of 49) were left language-dominant, as determined by Wada test. Two participants were right hemisphere-dominant, and one had bilateral language dominance. The remaining 11 participants were not evaluated for language dominance; 9 of them were right-handed and thus were assumed left language-dominant for the purposes of the analysis of lateralization (see below). The participant with bilateral dominance, and the remaining 2 participants who did not undergo Wada test and who were left-handed were not included in the analysis of hemispheric asymmetry in Fig 6. All participants underwent audiological and neuropsychological assessment prior to electrode implantation, and none had auditory or cognitive deficits that would impact the results of this study. The participants were tapered off their antiepileptic drugs during chronic monitoring when RS data were collected.

## Experimental procedures

### Preimplantation neuroimaging

All participants underwent whole-brain high-resolution T1-weighted structural MRI scans before electrode implantation. In a subset of 10 participants (S2 Table), RS-fMRI data were used for estimates of functional connectivity. The scanner was a 3T GE Discovery MR750W with a 32-channel head coil. The pre-electrode implantation anatomical T1 scan (3D FSPGR BRAVO sequence) was obtained with the following parameters: FOV = 25.6 cm, flip angle = 12 deg., TR = 8.50 ms, TE = 3.29 ms, inversion time = 450 ms, voxel size = $1.0 \times 1.0 \times 0.8$ mm. For RS-fMRI, 5 blocks of 5-minute gradient-echo EPI runs (650 volumes) were collected with the following parameters: FOV = 22.0 cm, TR = 2260 ms, TE = 30 ms, flip angle = 80 deg., voxel size = $3.45 \times 3.45 \times 4.0$ mm. In some cases, fewer RS acquisition sequences were used in the final analysis due to movement artifact or because the full scanning session was not completed. For each participant, RS-fMRI runs were acquired in the same session but noncontiguously (dispersed within an imaging session to avoid habituation). Participants were asked to keep their eyes open, and a fixation cross was presented through a projector.

### iEEG recordings

iEEG recordings were obtained using either subdural and depth electrodes or depth electrodes alone, based on clinical indications. Electrode arrays were manufactured by Ad-Tech Medical (Racine, WI). Subdural arrays, implanted in 36 participants out of 49, consisted of platinum-iridium discs (2.3 mm diameter, 5 to 10 mm inter-electrode distance), embedded in a silicon membrane. Stereotactically implanted depth arrays included between 4 and 12 cylindrical contacts along the electrode shaft, with 5 to 10 mm inter-electrode distance. A subgaleal electrode, placed over the cranial vertex near midline, was used as a reference in all participants. All electrodes were placed solely on the basis of clinical requirements, as determined by the team of epileptologists and neurosurgeons [151].

No-task RS data were recorded in the dedicated, electrically shielded suite in The University of Iowa Clinical Research Unit while the participants lay in the hospital bed. RS data were collected 6.4 +/− 3.5 days (mean ± standard deviation; range 1.5 to 20.9) after electrode implantation surgery. In the first 15 participants (L275 through L362), data were recorded using a TDT RZ2 real-time processor (Tucker-Davis Technologies, Alachua, FL). In the remaining 34 participants (R369 through L634), data acquisition was performed using a Neuralynx Atlas System (Neuralynx, Bozeman, MT). Recorded data were amplified, filtered (0.7 to 800 Hz bandpass, 5

dB/octave rolloff for TDT-recorded data; 0.1 to 500 Hz bandpass, 12 dB/octave rolloff for Neuralynx-recorded data), and digitized at a sampling rate of 2,034.5 Hz (TDT) or 2,000 Hz (Neuralynx). In all but 2 participants, recording durations were between 10 and 18 minutes, the median was 10; in 1 participant, duration was 6 minutes, and in 1 participant, the duration was 81 minutes.

## Data analysis

### Anatomical reconstruction and ROI parcellation

Localization of recording sites and their assignment to ROIs relied on post-implantation T1-weighted anatomical MRI and post-implantation computed tomography (CT). All images were initially aligned with preoperative T1 scans using linear coregistration implemented in FSL (FLIRT) [152]. Electrodes were identified in the post-implantation MRI as magnetic susceptibility artifacts and in the CT as metallic hyperdensities. Electrode locations were further refined within the space of the preoperative MRI using three-dimensional nonlinear thin-plate spline warping [153], which corrected for postoperative brain shift and distortion. The warping was constrained with 50 to 100 control points, manually selected throughout the brain, which were visually aligned to landmarks in the pre- and post-implantation MRI.

Electrode locations were mapped into a common anatomical template space using a combination of surface-based and volumetric coregistration. Automated identification and parcellation of the cortical surface within T1-weighted images were carried out with FreeSurfer [154,155]. Electrodes were assigned anatomical labels within the parcellation scheme of Destrieux and colleagues [156,157], according to the label of the nearest vertex (within the T1 image space) of the cortical surface mesh generated by FreeSurfer. Labeling was visually inspected and corrected whenever the automated parcellation did not conform to expected gyral boundaries. Volumetric mapping of T1 images to the MNI-152 space relied on automated linear coregistration implemented in the fsl_anat pipeline of the FSL toolbox [158]. Electrode coordinates in MNI-152 space were obtained by applying the resulting transformation to the coordinates from the T1 image space. The MNI coregistration was verified with a visual comparison of the transformed T1 with the template brain. Importantly, the MNI coordinates were only used for plotting the recording sites from multiple participants (Fig 1) to illustrate the overall extent of electrode coverage. MNI coordinates were not used for assigning sites to ROIs, nor were they used in any analyses presented in the study.

To pool data across participants, the dimensionality of connectivity matrices was reduced by assigning electrodes to one of 58 ROIs organized into 6 ROI groups (see Fig 1; S2 and S3 Tables) based upon anatomical reconstructions of electrode locations in each participant. For subdural arrays, ROI assignment was informed by automated parcellation of cortical gyri [156,157] as implemented in the FreeSurfer software package. For depth arrays, ROI assignment was informed by MRI sections along sagittal, coronal, and axial planes. For recording sites in Heschl's gyrus, delineation of the border between core auditory cortex and adjacent non-core areas (HGPM and HGAL, respectively) was performed in each participant using physiological criteria [159,160]. Specifically, recording sites were assigned to HGPM if they exhibited phase-locked (frequency-following) responses to 100 Hz click trains and if the averaged evoked potentials to these stimuli featured short latency ($<$20 ms) peaks. Such response features are characteristic for HGPM and are not present within HGAL [159]. Additionally, correlation coefficients between average evoked potential waveforms recorded from adjacent sites were examined to identify discontinuities in response profiles along Heschl's gyrus that could be interpreted as reflecting a transition from HGPM to HGAL. STG was subdivided into posterior and middle non-core auditory cortex ROIs (STGP and STGM), and auditory-related

anterior ROI (STGA) using the transverse temporal sulcus and ascending ramus of the Sylvian fissure as macroanatomical boundaries. The insula was subdivided into posterior and anterior ROIs, with the former considered within the auditory-related ROI group [32]. Middle and inferior temporal gyrus were each divided into posterior, middle, and anterior ROIs by diving the gyrus into 3 approximately equal-length thirds. Angular gyrus was divided into posterior and anterior ROIs using the angular sulcus as a macroanatomical boundary. Anterior cingulate cortex was identified by automatic parcellation in FreeSurfer and was considered as part of the prefrontal ROI group, separately from the rest of the cingulate gyrus. Postcentral and precentral gyri were each divided into ventral and dorsal portions using the $z_{MNI}$ coordinate (see below) of 40 mm as a boundary. Recording sites identified as seizure foci or characterized by excessive noise, or outside brain, were excluded from analyses and are not listed in S2 Table. Depth electrode contacts localized to the white matter were also excluded. Location within cortical white matter was determined based on visual inspection of anatomical reconstruction data (MRI sections along sagittal, coronal, and axial planes) as done in our previous studies (e.g., [62]). Electrode coverage was largely restricted to a single hemisphere in individual participants, and contacts on the contralateral hemisphere were excluded from analysis (and are not listed in S2 Table) such that all connections represent intrahemisphere functional connectivity.

## Preprocessing of fMRI data

Standard preprocessing was applied to the RS-fMRI data acquired in the pre-implantation scan using FSL's FEAT pipeline, including spatial alignment and nuisance regression. White matter, cerebrospinal fluid, and global ROIs were created using deep white matter, lateral ventricles, and a whole brain mask, respectively. Regression was performed using the time series of these 3 nuisance ROIs as well as 6 motion parameters (3 rotations and 3 translations) and their derivatives, detrended with second-order polynomials. Temporal bandpass filtering was 0.008 to 0.08 Hz. Spatial smoothing was applied with a Gaussian kernel (6 mm full-width at half maximum). The first 2 images from each run were discarded. Frame censoring was applied when the Euclidean norm of derivatives of motion parameters exceeded 0.5 mm [161]. All runs were processed in native EPI space and then the residual data were transformed to MNI152 and concatenated.

## Preprocessing of iEEG data

Analysis of iEEG data was performed using custom software written in MATLAB Version 2020a programming environment (MathWorks, Natick, MA, USA). After initial rejection of recording sites identified as seizure foci, several automated steps were taken to exclude recording channels and time intervals contaminated by noise. First, channels were excluded if average power in any frequency band [broadband, delta (1 to 4 Hz), theta (4 to 8 Hz), alpha (8 to 13Hz), beta (13 to 30 Hz), gamma (30 to 50 Hz), or high gamma (70 to 110 Hz); see below] exceeded 3.5 standard deviations of the average power across all channels for that participant. Next, transient artifacts were detected by identifying voltage deflections exceeding 10 standard deviations on a given channel. A time window was identified extending before and after the detected artifact until the voltage returned to the zero-mean baseline plus an additional 100 ms buffer before and after. High-frequency artifacts were also removed by masking segments of data with high gamma power exceeding 5 standard deviations of the mean across all segments. Only time bins free of these artifact masks were considered in subsequent analyses. Artifact rejection was applied across all channels simultaneously so that all connectivity measures were derived from the same time windows. Occasionally, particular channels survived the initial

average power criteria yet had frequent artifacts that led to loss of data across all the other channels. There is a tradeoff in rejecting artifacts (losing time across all channels) and rejecting channels (losing all data for that channel). If artifacts occur on many channels, there is little benefit to excluding any one channel. However, if frequent artifacts occur on one or simultaneously on up to a few channels, omitting these can save more data from other channels than those channels contribute at all other times. We chose to optimize the total data retained, channels × time windows, and omitted some channels when necessary.

On occasion, noise from in-room clinical equipment and muscle artifacts appeared in the data as shared signals across channels. These types of noise were typically broadband and could be detected via analysis of frequencies higher than those of interest here. To remove these signals, data from retained channels were high-pass filtered above 200 Hz, and a spatial filter was derived from the singular value decomposition omitting the first singular vector. This spatial filter was then applied to the broadband signal to remove the common signal. This procedure was implemented as follows.

For the array $\mathbf{y}_{\mathrm{HP}}$ of $N$ samples of high-pass filtered data from $M$ recording sites, and the covariance matrix $C_{M \times M} = \mathbf{S}(\mathbf{y}_{HP}^T \mathbf{y}_{HP})\mathbf{S}$, where

$$\mathbf{S} = \begin{bmatrix} 1/\sigma_1 & 0 & \dots \\ 0 & \ddots & \vdots \\ \vdots & 0 & 1/\sigma_M \end{bmatrix},$$

and $\sigma_i$ = standard deviation of high-passed filtered signal from the $i^{\text{th}}$ recording site, $y_{\mathrm{HP,i}}$, the singular value decomposition of $C_{M \times M}$ is obtained as

$$C_{M \times M} = [\mathbf{u}_1, \mathbf{u}_2, \dots, \mathbf{u}_M] \begin{bmatrix} \lambda_1 & 0 & \dots \\ 0 & \ddots & 0 \\ \vdots & 0 & \lambda_M \end{bmatrix} [\mathbf{u}_1, \mathbf{u}_2, \dots, \mathbf{u}_M]^T,$$

where $\mathbf{u}_i$ are eigenvectors, and $\lambda_i$ are eigenvalues of $C_{M \times M}$. The spatial filter is defined as

$$\mathbf{W}_{SVD} = \mathbf{S}(I_{M \times M} - \mathbf{u}_1 \mathbf{u}_1^T)\mathbf{S}^{-1},$$

where $I_{\mathrm{M \times M}}$ is the identity matrix. The spatial filter is applied to the unfiltered data, $\mathbf{y}$, as

$$\mathbf{y}_{\mathrm{SVD}} = \mathbf{y} \mathbf{W}_{\mathbf{SVD}}.$$

The purpose of the initial high-pass filtering in computing the spatial filter, $\mathbf{W}_{SVD}$, is to minimize the influence of long-range physiological correlations, which tend to be associated with low frequency oscillations [162], on $C_{M \times M}$, while preserving zero-lag correlations arising from reference contamination and other potential artifactual sources.

## Connectivity analysis

For RS-fMRI data, BOLD signals were averaged across voxel groupings and functional connectivity was calculated as Pearson correlation coefficients. Voxel groupings were either based on the Schaefer–Yeo 400 parcellation scheme [90] in MNI-152 space or were based on iEEG electrode location in participant space (see Fig 1). For the latter, fMRI voxels were chosen to represent comparable regions of the brain recorded by iEEG electrodes. For each electrode, the anatomical coordinates of the recording site were mapped to the closest valid MRI voxel, $E$, and a sphere of 25 voxels (25 mm$^3$) centered on $E$ used as the corresponding recording site.

This process was repeated for all *N* electrodes in the same ROI, and a single time series computed as the average of the fMRI BOLD signal in these $N \times 25$ voxels. These averages were used to compute an ROI-by-ROI connectivity matrix for RS-fMRI data. For comparisons between iEEG and fMRI embeddings, voxels were processed in participant space and ROI labels from the parcellation scheme illustrated in Fig 1 and S2 Table were applied to the fMRI data. For comparisons between fMRI embeddings derived from all cortical ROIs versus fMRI embeddings derived from just ROIs sampled in the iEEG experiments, electrode locations were transformed from participant space to MNI-152 space and then assigned to ROIs within the Schaefer–Yeo 400 scheme.

Connectivity was measured for iEEG data using orthogonalized band power envelope correlation [68]. This measure avoids artifacts due to volume conduction by discounting connectivity near zero phase lag. Data were divided into 60-second segments, pairwise connectivity estimated in each segment, and then connectivity estimates averaged across all segments for that subject. Power envelope correlations were calculated using a method similar to [68], except time-frequency decomposition was performed using the demodulated band transform [163] rather than wavelets. For each frequency band (theta, alpha, beta, gamma; high gamma), the power at each time bin was calculated as the average (across frequencies) log of the squared amplitude. For each pair of signals *X* and *Y*, one was orthogonalized to the other by taking the magnitude of the imaginary component of the product of one signal with the normalized complex conjugate of the other:

$$Y_{orth} = |\mathrm{Im}\{Y \times X^*/|X|\}|$$

Both signals were bandpass filtered (0.2 to 1 Hz), and the Pearson correlation calculated between signals. The process was repeated by orthogonalizing in the other direction and the overall envelope correlation for a pair of recording sites was the average of the 2 Pearson correlations. Lastly, correlations were averaged across segments.

Results for envelope correlations were compared to those derived from the debiased wPLI [70], a measure of phase synchronization. This measure also avoids artifacts due to volume conduction by discounting connectivity near zero phase lag. wPLI was estimated for each 60-second data segment and every recording site pair from the sign of the imaginary part of the cross-spectrum at each frequency and averaged across frequencies within each band of interest (theta: 4 to 8 Hz, alpha: 8 to 13 Hz, beta: 13 to 30 Hz). The cross spectrum was calculated from the demodulated band transform as described previously [62].

Connectivity matrices were thresholded prior to DME to reduce the contribution of spurious connections to the analysis. We balanced our desire to minimize noisy connections while maintaining a connected graph (i.e., that that there are no isolated nodes; required by DME [59]) by saving at least the top third (rounded up) connections for every row, as well as their corresponding columns (to preserve symmetry). To ensure that the graph was connected after thresholding, we also included any connections making up the minimum spanning tree of the graph represented by the elementwise reciprocal of the connectivity matrix to ensure the graph is connected.

## ROI-based connectivity analysis

Connectivity between ROIs was computed as the average envelope correlation between all pairs of recording sites in the 2 ROIs. For analyses in which connectivity was summarized across participants (Figs 3–8), we used only a subset of ROIs such that every possible pair of included ROIs was represented in at least 2 participants (S2 Table). This list of ROIs was obtained by iteratively removing ROIs with the worst cross-coverage with other ROIs until every ROI remaining had sufficient coverage with all remaining ROIs.

## Diffusion map embedding

See the S1 Text for details about DME. In brief, the functional connectivity is transformed by applying cosine similarity [25] to yield the similarity matrix $\mathbf{K} = [k(i,j)]$. This matrix is then normalized by degree to yield a matrix $\mathbf{P} = \mathbf{D}^{-1}\mathbf{K}$, where $\mathbf{D}$ is the degree matrix, i.e., the diagonal elements of $\mathbf{D} = \sum_{j=1}^{M} k(i,j)$, where $M$ is the number of recording sites, and the off-diagonal elements of $\mathbf{D}$ are zero. If the recording sites are conceptualized as nodes on a graph with edges defined by $\mathbf{K}$, then $\mathbf{P}$ can be understood as the transition probability matrix for a "random walk" or a "diffusion" on the graph (see S1 Text; [59,60]). DME consists of mapping the recording sites into an embedding space using an eigendecomposition of $\mathbf{P}$,

$$\Psi^{(t)}(x_i) = [\lambda_1^{t}\psi_1(x_i), \lambda_2^{t}\psi_2(x_i), \ldots, \lambda_M^{t}\psi_M(x_i)]^T$$

where $\psi_j$ are the eigenvectors of $\mathbf{P}$.

The parameter $t$ corresponds to the number of steps in the diffusion process (random walk on the graph). The coordinates of the data in embedding space are scaled according to $\lambda_i^{t}$, where $\lambda_i$ is the eigenvalue of the $i^{th}$ dimension being scaled. Thus, the value of $t$ sets the spatial scale of the analysis, with higher values de-emphasizing smaller eigenvalues. Because $|\lambda_i| < 1 \,\forall\, i$, at higher values of $t$ each dimension will be scaled down ('collapse'), with the dimension corresponding to $\max(|\lambda_i|)$ (i.e., $\lambda_1$) scaled the least. A "multiscale" DME analysis, in which all values of $t$ are considered simultaneously (and thus the analysis no longer depends on a specific values of $t$) has been implemented as well [69]. Because we wished to explore the structure of the data over multiple dimensions of embedding space, we restricted our analyses to smaller values of $t$. Here, we present data for $t = 1$ but compare our key results to those obtained using multiscale DME.

DME can be implemented alternatively based on a symmetric version of diffusion matrix $\mathbf{P_{symm}} = \mathbf{D}^{-0.5}\mathbf{K}\mathbf{D}^{-0.5}$. Basing DME on $\mathbf{P_{symm}}$ has the advantage that the eigenvectors of $\mathbf{P_{symm}}$ form an orthogonal basis set (unlike the eigenvectors of $\mathbf{P}$), providing some additional convenience mathematically that is beyond the scope of this paper [60]. Additionally, the eigenvalues of $\mathbf{P}$ and $\mathbf{P_{symm}}$ are identical.

In 2 sets of analyses presented here, pairs of embeddings were compared to each other: in the analysis of lateralization of speech and language networks, and in the comparison between iEEG and fMRI data. To do that, we used a change of basis operator to map embeddings into a common embedding space using the method described in Coifman and colleagues [60].

## Dimensionality reduction via low rank approximations to $\mathbf{P_{symm}}$

DME offers an opportunity to reduce the dimensionality of the underlying data by considering only those dimensions that contribute importantly to the structure of the data, as manifested in the structure of the transition probability matrix $\mathbf{P}$, or, equivalently, of the diffusion matrix $\mathbf{P_{symm}}$. We used the eigenvalue spectrum of $\mathbf{P_{symm}}$ to determine its ideal low rank approximation, balancing dimensionality reduction and information loss. The basis for this is most easily understood in terms of the eigenvalue spectrum of $\mathbf{P}$, whose spectrum is identical to that of $\mathbf{P_{symm}}$ [60]. Because $\mathbf{P}$ is real and symmetric, the magnitude of the eigenvalues is identical to the singular values of $\mathbf{P}$. The singular values tell us about the fidelity of low rank approximations to $\mathbf{P}$. Specifically, if $\mathbf{P}$ has a set of singular values $\sigma_1 \geq \sigma_2 \geq \ldots \geq \sigma_n$, then for any integer $k \geq 1$,

$$\min_{\tilde{\mathbf{P}}_k} \|\mathbf{P} - \tilde{\mathbf{P}}_k\|_2 = \sigma_{k+1},$$

where $\tilde{\mathbf{P}}_k$ is the rank-$k$ approximation to $\mathbf{P}$. Thus, the magnitude of the eigenvalues corresponds to the fidelity of the lower dimensional approximation, and the difference in the magnitude of

successive eigenvalues represents the improvement in that approximation as the dimensionality increases. The spectrum of $\mathbf{P}$ invariably has an inflection point ("elbow") at $i = k_{\text{infl}}$, separating 2 sets of eigenvalues $\lambda_i$: those whose magnitude decreases more quickly with increasing $i$ until the inflection point, and those beyond the inflection point whose magnitude decreases more slowly with increasing $i$. The inflection point thus delineates the number of dimensions that are most important for approximating $\mathbf{P}$ or $\mathbf{P_{symm}}$. The inflection point $k_{\text{infl}}$ was identified algorithmically [164], and the number of dimensions retained set equal to $k_{\text{infl}}- 1$.

## Comparing distances in embedding space

The relative distance between points in embedding space provides insight into the underlying functional geometry. In several analyses presented here, 2 embeddings of identical sets of ROIs were compared as ROI distances within the 2 embeddings. After mapping to a common space and reducing dimensionality as described above, the 2 embeddings A and B were used to create the pairwise distance matrices A'and B'. The Pearson correlation coefficient $r$ was then computed between the upper triangles (excluding the diagonal) of the corresponding elements in the distance matrices. To compare anatomical distance and distance in embedding space, inter-ROI anatomical distances were calculated for each participant by computing the centroid of each ROI in MNI space, then calculating Euclidean distances between centroids, followed by averaging distances across participants.

## Signal-to-noise (SNR) characteristics

To measure the robustness of the embedding analysis to variability over time, an SNR was computed as follows. For each participant, a channel × channel $\mathbf{P_{symm}}$ matrix was calculated for each 60-second segment of data. For each segment, DME analysis was applied and a channel × channel distance matrix calculated. These distance matrices were averaged across segments. The "signal" of interest was defined as the variability (standard deviation) of this averaged distance matrix (ignoring the diagonals). The "noise" was defined as the variability across time, estimated for each element of the distance matrix as the standard deviation across segments, then averaged across the elements of the matrix. The SNR for functional connectivity itself was computed in an analogous manner, using the original channel × channel connectivity matrix rather than the matrix of embedding distances.

## Estimating precision in position and distances in embedding space

To obtain error estimates for both ROI locations in embedding space and embedding distance between ROIs, average ROI × ROI adjacency matrices were calculated. Our data are hierarchical/multilevel, in that we sampled participants in whom there are multiple recording sites. Nonparametric bootstrap sampling at the highest level ("cluster bootstrap" [165]; here, the word "cluster" refers to the hierarchical/multilevel structure of the data, with multiple recording sites within participants, rather than algorithmic clustering) is the preferred approach for hierarchical data when groups (here, participants) are sampled and observations (here, recording sites) occur within those groups [166], with fewer necessary assumptions than multilevel (mixed-effects) modelling (e.g., subject effects are not assumed to be linear). Using this approach, participants were resampled with replacement, connectivity averaged across the bootstrapped samples, and DME performed for 100,000 such adjacency matrices. For locations in embedding space, these embeddings were then mapped via the change of basis procedure described above to the original group average embedding space. For each ROI, the mapped bootstrap iterations produced a cloud of locations in embedding space that were summarized by the standard deviation in each dimension. For embedding distances, no change of basis was

necessary because distances are preserved across bases. As the bootstrapping procedure was applied at the participant level, the variance estimates are most comparable to those of a model accounting for random effects of participant, i.e., they represent estimates accounting for participant-level variation in the sample rather than recording site-level estimates, with or without correcting for correlations within levels.

To compare the positions of STSL versus STSU relative to canonical auditory cortical ROIs (HGPM, HGAL, PT, PP, STGP, and STGM) or ROIs involved in semantic processing (STGA, MTGA, MTGP, ITGA, ITGP, TP, AGA, AGP, SMG, IFGop, IFGtr, IFGor; [21,78–80]), we calculated the average pairwise distance from STSL or STSU to each such ROI. The difference between these averages was compared to a null distribution obtained by Monte Carlo sampling of the equivalent statistic obtained by randomly exchanging STSL/STSU labels by participant. The specific comparisons performed were chosen a priori to constrain the number of possible hypotheses to test; pairwise comparisons of all possible ROI pairs (let alone comparisons of all higher-order groupings) would not have had sufficient statistical power under appropriate corrections for multiple comparisons. Though different choices could have been made for inclusion in the "semantic processing" category, exchanging 1 or 2 of these ROIs would not strongly influence the average distance in a group of 12 ROIs.

## Hierarchical clustering

Agglomerative hierarchical clustering was done using the *linkage* function in MATLAB, with Euclidean distance as the distance metric and Ward's linkage (minimum variance algorithm) as the linkage method. The ordering of ROIs along the vertical axis in the dendrogram was determined using the *optimalleaforder* function in MATLAB, with the optimization criterion set to "group."

Nonparametric bootstrapping at the participant level, as described above, was used to evaluate the robustness of clustering results both overall and at the level of individual clusters. We compared the original cluster results obtained with the full dataset to the result obtained with each bootstrap sample and then summarized those results across iterations.

Overall stability of the cluster results was evaluated using the median normalized Fowlkes–Mallows index [85] across cluster bootstrap iterations, noted as $B_k$ for $k$ clusters (see S2 Text). Normalizing to the expected value $E(B_k)$ (see S2 Text) results in an index where 0 represents average random (chance) clustering and 1 represents perfectly identical clustering.

Cluster-wise stability was calculated by the membership of each cluster at each iteration to the corresponding cluster obtained with the full dataset using the maximum Jaccard coefficient for each reference cluster [86]. The Jaccard coefficient varies from 0 (no overlap in cluster membership) to 1 (identical membership) and is defined as the ratio of the size of the set containing intersection of the 2 clusters divided by the size of the set containing their union. We then subtracted from this coefficient a bias estimate calculated by randomly permuting the cluster assignments on each bootstrap iteration.

## Auditory responsiveness

In a subset of 37 participants, auditory responsiveness was evaluated as percentage of sites within each ROI that exhibited high gamma responses to monosyllabic word stimuli. The stimuli were monosyllabic words ("cat," "dog," "five," "ten," "red," "white"), obtained from TIMIT (https://doi.org/10.35111/17gk-bn40) and LibriVox (http://librivox.org/) databases. The words were presented in semantic categorization (animals and numbers target categories) and tone target detection tasks as described previously [23,87–89]. A total of 20 unique exemplars of each word were presented in each task: 14 spoken by different male and 6 by different

female speakers. The stimuli were delivered via insert earphones (ER4B, Etymotic Research, Elk Grove Village, IL) integrated into custom-fit earmolds. All stimuli had a duration of 300 ms, were root-mean-square amplitude-normalized, and were delivered in random order. The interstimulus interval was chosen randomly within a Gaussian distribution (mean 2 s; SD = 10 ms). The task was to push a response button whenever the participant heard a target sound. The hand ipsilateral to the hemisphere in which the majority of electrodes were implanted was used to make the behavioral response. There was no visual component to the task, and the participants did not receive any specific instructions other than to respond to target auditory stimuli by pressing a button. Mean high gamma (70 to 110 Hz) power within early (50 to 350 ms) and late (350 to 650 ms) poststimulus time windows was compared with that in a prestimulus window (−200 to −100 ms). Both nontarget and target trials were included in the analysis to maximize its sensitivity. Significance of high gamma responses was established at a $\alpha$ = 0.05 level using one-tailed Mann–Whitney $U$ tests with FDR correction.

## Comparing language dominant/nondominant hemispheres

To test for differences in functional geometry between language dominant and nondominant hemispheres, 2 measures were considered: differences in the location of individual ROIs in embedding space, and different pairwise distances between ROIs in embedding space. To calculate differences in location of individual ROIs, dominant/nondominant average embeddings were mapped to a common space (from an embedding using the average across all participants regardless of language dominance) using the change of basis operator. The language-dominant location difference for a specific ROI was calculated as the Euclidean distance between the 2 locations of each ROI in this common space. To examine whether there was a consistent relationship between hemispheric asymmetry in a given ROI's location in embedding space and the percentage of either early or late auditory responsive sites within that ROI, two-tailed Spearman's rank tests were used. To calculate differences in pairwise distances between ROIs, Euclidean distances were calculated in embedding space for each hemisphere and then subtracted to obtain a difference matrix. To determine whether the differences in location or pairwise distances were larger than expected by chance, random permutations of the dominant/nondominant labels were used to generate empirical null distributions. Since this approach produces a $p$-value for every pair of connections, $p$-values were adjusted using FDR to account for multiple comparisons.

## Analyses of fMRI connectivity in embedding space

Two sets of analyses were performed using fMRI data. First, iEEG and fMRI data were compared in embedding space. In this analysis, connectivity based on RS-fMRI data from voxels located at electrode recording sites was compare with the corresponding connectivity matrix derived from iEEG data. The embedding analysis was applied to the 2 connectivity matrices, all pairwise inter-ROI distances computed, and iEEG and fMRI data compared using the correlation of the pairwise ROI distances. The second analysis was to compare embeddings derived from all ROIs in the RS-fMRI scans to those derived from just ROIs sampled with iEEG electrodes. Here, ROI × ROI connectivity matrices were computed for all ROIs, then embeddings created from the full matrices or from matrices containing just rows and columns corresponding to the ROIs sampled with iEEG.

## Supporting information

**S1 Fig. Embedding plots in participant R376 plotted on the same scale in 4 dimensions.** (TIF)

**S2 Fig. Comparison of signal-to-noise ratio (SNR) for the embedding analysis versus direct analysis of functional connectivity.** Each symbol corresponds to 1 participant. For each participant, the SNRs of embedding distances and connectivity were calculated from the recorded RS block as described in Methods. In most participants, the embedding analysis exhibited superior SNR characteristics compared to direct analysis of connectivity.
(TIF)

**S3 Fig. Average gamma band embedding, plotted on the same scale in the first 4 dimensions.**
(TIF)

**S4 Fig. Average beta band embedding, plotted on the same scale in the first 4 dimensions.**
(TIF)

**S5 Fig. Sensitivity of DME results to the choice of threshold, frequency band, and connectivity measure.** (**a**) Comparison of embedding results across thresholds applied to connectivity matrices. Threshold = 33% was used for the results of the main figures. (**b**) Comparison of embedding results across frequency bands and functional connectivity measures. Threshold = 33%. For (**a**) and (**b**), data shown are Pearson correlations of inter-ROI distances from embeddings obtained with the different thresholds, measures, and bands.
(TIF)

**S6 Fig. Hierarchical clustering of data in embedding space for all studied bands.** (**a**) theta; (**b**) alpha; (**c**) beta; (**d**) gamma (same data as in Fig 4); (**e**) high gamma. Linkages between ROI groups identified using agglomerative clustering. As in Fig 4, 2 thresholds are shown for each band, $n_{Cluster}$ = 5 and 9 (vertical dashed lines). The number of clusters is the number of lines in the dendrogram intersected by the threshold line. Clusters consist of all ROIs descending from the intersected line. The color scheme for ROI labels is set by the gamma parcellation.
(TIF)

**S7 Fig. Sensitivity of hierarchical clustering results to threshold and choice of diffusion parameter $t$.** (**a-e**) Effect of varying threshold from 10%–50% with $t$ = 1. (**f**) Clustering results for embeddings derived using the multiscale approach instead of $t$ = 1. For all panels, linkages between ROI groups were identified using agglomerative clustering. As in Fig 4, 2 thresholds are shown, $n_{Cluster}$ = 5 and 9 (vertical dashed lines). The number of clusters is the number of lines in the dendrogram intersected by the threshold line. Clusters consist of all ROIs descending from the intersected line. The color scheme for ROI labels is set by the gamma parcellation with threshold 33% and $t$ = 1.
(TIF)

**S8 Fig. Stability of cluster results.** (**a**) Stability of overall cluster results shown in Fig 4A and S6 Fig was evaluated for each frequency band as a function cluster number using the Fowlkes–Mallows score. (**b**) Cluster-wise stability for gamma band data as a function of cluster number was evaluated using the Jaccard index.
(TIF)

**S9 Fig. Auditory networks do not differ between hemispheres.** Data plotted on the same scale in the first 4 dimensions of embedding space for all dominant and nondominant participants (**a**), just dominant (**b**), and just nondominant (**c**).
(TIF)

**S1 Movie. Average gamma band embedding, dimensions 2–4.**
(MP4)

**S2 Movie. Average gamma band embedding, dimensions 3–5.**
(MP4)

**S1 Table. Participant demographics and study information.**
(XLSX)

**S2 Table. ROIs and electrode coverage.** Participant and site numbers that were not included in group averages are denoted in gray italics.
(XLSX)

**S3 Table. List of abbreviations.**
(XLSX)

**S1 Text. Diffusion map embedding details.**
(DOCX)

**S2 Text. Fowkles–Mallows index.**
(DOCX)

# Acknowledgments

We are grateful to Jess Banks, Alex Billig, Haiming Chen, Phillip Gander, Christopher Garcia, Matthew Howard, Ariane Rhone, and Matthew Sutterer for help with data collection, analysis, and comments on the manuscript.

# Author Contributions

**Conceptualization:** Matthew I. Banks, Kirill V. Nourski.

**Data curation:** Bryan M. Krause, Joel E. Bruss, Christopher K. Kovach, Hiroto Kawasaki, Kirill V. Nourski.

**Formal analysis:** Matthew I. Banks, Bryan M. Krause, D. Graham Berger, Declan I. Campbell, Kirill V. Nourski.

**Funding acquisition:** Matthew I. Banks, Kirill V. Nourski.

**Investigation:** Hiroto Kawasaki, Kirill V. Nourski.

**Methodology:** Matthew I. Banks, Bryan M. Krause, Aaron D. Boes, Joel E. Bruss, Christopher K. Kovach, Mitchell Steinschneider, Kirill V. Nourski.

**Project administration:** Kirill V. Nourski.

**Software:** Matthew I. Banks, Bryan M. Krause, D. Graham Berger, Declan I. Campbell, Christopher K. Kovach.

**Supervision:** Matthew I. Banks, Kirill V. Nourski.

**Validation:** Bryan M. Krause.

**Visualization:** Bryan M. Krause, Kirill V. Nourski.

**Writing – original draft:** Matthew I. Banks, Bryan M. Krause, Kirill V. Nourski.

**Writing – review & editing:** Matthew I. Banks, Bryan M. Krause, D. Graham Berger, Declan I. Campbell, Aaron D. Boes, Joel E. Bruss, Christopher K. Kovach, Mitchell Steinschneider, Kirill V. Nourski.

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
