## [Editor Report · Decision Letter 0]

7 Dec 2022

Dear Dr Banks, 

Thank you for submitting your manuscript entitled "Functional geometry of auditory cortical resting state networks derived from intracranial electrophysiology" for consideration as a Research Article by PLOS Biology.

Your manuscript has now been evaluated by the PLOS Biology editorial staff, as well as by an academic editor with relevant expertise, and I am writing to let you know that we would like to send your submission out for external peer review.

However, before we can send your manuscript to reviewers, we need you to complete your submission by providing the metadata that is required for full assessment (including things like suggesting possible reviewers, asking us to respect up to 3 excluded reviewers, conflicts of interest, etc). To this end, please login to Editorial Manager where you will find the paper in the 'Submissions Needing Revisions' folder on your homepage. Please click 'Revise Submission' from the Action Links and complete all additional questions in the submission questionnaire.

Once your full submission is complete, your paper will undergo a series of checks in preparation for peer review. After your manuscript has passed the checks it will be sent out for review. To provide the metadata for your submission, please Login to Editorial Manager (https://www.editorialmanager.com/pbiology) within two working days, i.e. by Dec 09 2022 11:59PM.

Kind regards,

Kris

Kris Dickson, Ph.D., (she/her)

Neurosciences Senior Editor/Section Manager

PLOS Biology

kdickson@plos.org

---

## [Decision Letter · Decision Letter 1]

6 Feb 2023

Dear Dr Banks,

Thank you for your patience while your manuscript "Functional geometry of auditory cortical resting state networks derived from intracranial electrophysiology" was peer-reviewed at PLOS Biology. Your manuscript has been evaluated by the PLOS Biology editors, an Academic Editor with relevant expertise, and by several independent reviewers.

As you will see in the reviewer reports, which can be found at the end of this email, although the reviewers find the work potentially interesting, they have also raised a substantial number of important concerns. Based on their specific comments and following discussion with the Academic Editor, it is clear that a substantial amount of work would be required to meet the criteria for publication in PLOS Biology. However, given our and the Academic Editor's interest in your study, we would be open to inviting a comprehensive revision of the study that thoroughly and comprehensively addresses all the reviewers' comments. 

Given the extent of revision that would be needed, we cannot make a decision about publication until we have seen the revised manuscript and your response to the reviewers' comments. Your revised manuscript would need to be seen by the reviewers again, but please note that we would not engage them unless their main concerns have been addressed. We appreciate that these requests represent a great deal of extra work, and we are willing to relax our standard revision time to allow you 6 months to sufficiently revise your study. We also appreciate that, given the extent of revisions needed, you might prefer to consider submission elsewhere. At this stage, your manuscript remains formally under active consideration at our journal. Please do notify us by email if you do not intend to submit a revision so that we may withdraw it.

Please email us (plosbiology@plos.org) if you have any questions or concerns, or envision needing a (short) extension.

**IMPORTANT - SUBMITTING YOUR REVISION**

*Resubmission Checklist*

*Published Peer Review*

*PLOS Data Policy*

*Blot and Gel Data Policy*

Sincerely,

Kris

Kris Dickson, Ph.D., (she/her)

Neurosciences Senior Editor/Section Manager

PLOS Biology

kdickson@plos.org

REVIEWS:

Reviewer's Responses to Questions

Do you want your identity to be public for this peer review?

Reviewer #1: No

Reviewer #2: No

Reviewer #3: No

Reviewer #1: In this article, Banks and colleagues investigate the geometry of auditory resting state networks derived from iEEG. They used Diffusion Map Embedding to represent recording sites in a low-dimensional space. This approach permits the authors to characterize some specificities of the auditory networks at three scales. At the fine scale, they show a segregation of auditory cortical sites. At the mesoscale, they report the cluster of a limbic auditory stream. At the macroscale, they found a greater heterogeneity for the language dominant hemisphere.

The innovative approach followed in this study is interesting and brings pertinent insights on the current states of the art of human brain network organization at multiple levels. Please find below my comments to improve the impact of the paper and alleviate potential issues.

Majors:

(1) There are two important aspects of the study that might be criticized by the community: the data used for the analyses correspond to resting state data, and while the main analyses rely on functional connectivity the link with brain function is somehow indirect.

Resting state data does not correspond to (pro)active processing performed by the brain. We know that neural networks are shaped by the type of computation, that is they can change depending on task demand (e.g., attention, memory etc.). The pertinence of resting state is still a matter of debate for some researchers, and it would be appropriate to tackle this possible limitation, at least in the discussion. The geometrical organization described in the study might be subjected to change if participants would have been asked to perform different active tasks. Therefore, when it comes to relate the present results to function some assertion could be tone down or clarified. For instance, the use of the word 'functional', 'functionally' or 'processing hierarchy' might lead to some misinterpretation as it related to the measure and not to brain function.

The authors tend to assess functionality by using additional dataset in a subset of participants, where mono-syllabic words were presented in semantic categorization and tone target detection tasks. Yet, this analysis and the corresponding results are limited: the authors do not provide enough details about the different data types, the processing was focused on high frequency activity and the topology of the network was not investigated during these tasks.

(2) The main analysis is conducted on orthogonalized power envelope correlation in the gamma-band.

Additional analyses are performed on other frequency bands using either the same metric, or wPLI (not fully described in the materials and methods section). Supplementary Figure 8 summarizes the correlation between the different approaches. Yet not much attention is brought on the differences, although the authors underlie its importance (lines 297-298, lines 325-326 or lines 549-551). For instance, the results in the beta band suggest the existence of different geometrical configurations (i.e., in the beta band the correlation with other bands is the lowest). Knowing that the beta band can reflect top-down process, as compared to bottom-up process possibly carried by lower and higher frequency bands, one may wonder if the geometry of auditory network changes with the type of process/task. Moreover, the different metrics (magnitude or phase) do not have the same neurophysiological meaning.

It would be critical for the community to approach these questions. Even if it is not the main goal of the study, measuring and reporting the malleability/rigidity of the geographical distances would certainly be beneficial for the impact of the paper.

(3) The authors use fMRI data to, somehow, validate their results. It brings the question of the benefit of using iEEG rather than fMRI. How to convince the community about the advantage of using iEEG (i.e., clinical data difficult to get) rather than fMRI if the latest is used as a form of control and easier to get?

The most immediate response would be the difference in time resolution. But this advantage is not obvious in the current state of the manuscript.

Also, it is known that the BOLD signal is positively correlated with iEEG gamma band activity and inversely correlated with alpha band activity. This might require more attention (e.g., in link with lines 406-408 or with Figure 7 panel d).

(4) The authors wrote (line 324) that the results are consistent with network organization depending on temporal scale. It would be pertinent to further develop this argumentation as it is in direct link with the main characteristic of iEEG, its high temporal resolution. Maybe clarifying the meaning behind parameter "t" would be important (in the main text not in the supplements). Making links with the works on brain temporal time scales might be relevant too.

(5) Somehow, this article 'straddles' between being a purely methodological paper and a study on brain network geometry. That is, it is not clear if this study is a proof of concept of a method (i.e., Diffusion Map Embedding) applied to iEEG or an unprecedented investigation of the organization of auditory brain network. These two aspects of the paper are not mutually exclusive, but maybe this could be further delineated throughout the manuscript.

Minors:

- In the introduction is mentioned the fact that investigation of the anterior temporal lobe with fMRI is subjected to methodological issues, but the ANT is not the focus of the present study.

- In the introduction, when different auditory areas are described the concepts of specificity and sensitivity should be clarified.

- The introduction finishes with a long sentence that is not easy to understand. Maybe it could be related to the results of the study.

- Information is lacking concerning the methods used to normalize data into the template(s) (e.g., MNI, atlases…).

- In Figure 1, the purplish colors for the Auditory core and sensori-motor regions are not easy to distinguish.

- The variance between participants should be reported (e.g., in the functional geometry of cortical networks).

- Can the authors add the comparisons between dimensions 2 & 3, 3 & 4 and 2 & 4?

- line 202, maybe the term "phase" could be replaced by 'time-locked' or nothing that would permit to avoid confusion with the phase of the signal which is not specifically analyzed here.

- What is the difference between the terms 'polysensory' (line 226), multimodal (line 248) or 'transmodal' (line 541, 548)? if they convey the same meaning, the same term must be used.

- The PP and InsP are not easily visible in Figure 3. Because those regions are discussed in detail, they could be somehow highlighted in the figure.

- line 251, it is said that the brain parcellation in Figure 1 is suboptimal because there are no quantitative criteria for designating 'auditory-related' and 'auditory non-core'. But these areas are color coded in green and yellow, respectively. Maybe this can be clarified?

- In Figure 4 panel b, to facilitate the comparison dashed lines extending the x-axis along the y-axis would be useful. In panel c, the colors coding is not enough contrasted to distinguish the different clusters.

- In supplementary Figure 5, it would be nice to also have the derived brain parcellation based on hierarchical clustering (like in Figure 4).

- lines 302-304: Actually, there are tools/metrics, from graph theory that define centrality and strong connectivity. The corresponding sentence should be revised accordingly.

- line 312-313 and Figure 5b, it is not clear how the distance from the center could not be inversely proportional to the connectivity metric used. How would the authors explain a scenario where it would not be the case (as one measure is derived from the other)?

- In Figure 5 panel b, it is not clear how the four dimensions are collapsed.

- In Figure 7 panel C, there is a cloud of points showing a horizontal trend, can the authors provide information about it?

- lines 460-461, what are the data-driven hypotheses for future studies?

- Some considerations about the visual and sensory-motor areas would nicely complement the discussion on FG and MOG on multisensory integration.

- line 662, the length of the data is really 13 +/- 11 min?

- How does electrodes in the white matter were identify?

- lines 740-744, the procedure used to avoid volume conduction is described without explaining how the use of high frequency activity can help to get rid of passive signal spread that dominate low frequencies.

- line 776, the rationale for using the top-third connections is missing.

- In supplementary table 1, please add the number of ECoG electrodes and sEEG electrodes per participants.

- In the legend of supplementary Figure 2, it seems that 'noise' should be SNR.

Reviewer #2: This is a very interesting report on a large data set of invasive (stereotactical EEG) recordings in human patients, using resting-state data to establish/understand better potential topological relations in a dimensionality-reduced network representation of these data (all being based on gamma-band power envelopes; see specific comments on this choice below).

The paper shares with many other topological analysis papers its pro's and con's: It goes far beyond other iEEG/intracranial studies in that it has larger coverage and larger N than many other studies before and thus is able to describe the data at a more abstract level. (but see statistical concerns on pooling data from participants below.) As a problem arising from this, it remains unclear (to me at least) what the achived level of description really adds beyond known anatomical hierarchical relations, in lieu of functional data and/or behavioural markers. The paper as such feels somewhat premature and the data might better serve as an atlas-like basis for future functional interrogations.

In sum, I was intruiged but ultimately not convinced about a new mechanistic insight being conveyed.

More detailed comments follow below, page numbers referring to the PDF made available by Plos Biol:

[page 22]: I overall struggled to see the deeper utility of the present data. I am not questioning the merit of iEEG with cortex-wide coverage for our progess in understanding cortical functional organisation. However, the present data use resting-state data to make in places somewhat ciurcular conclusions from the anatomical structures where electrodes were placed to the functions ascribed to the regions.

Clear new insights are sparse from these analyses, accordingly. Their value might lie more in a serving as a descriptive, atlas-like basis for future studies and the data should be made publicly available, accordingly.

[page 17]: Pooling across participants: To the best of my understanding, the analyses as reported do not take particular care of the hierarchical structure of these data. The network analyses seem to rest on pooled data across electrodes and participants.

This essentially equals what could be referred to as a "fixed effects" analysis.

It is unclear how a generalisation across electrodes/participants sampled is actually feasible here.

[page 23]: The rationale of working off gamma-band envelopes when building/analysing large-scale brain-wide networks remained somewhat elusive. The authors argue more in passing than anything (line 294f and supplements) that there was an overlap with networks derived from the (in my mind more plausible for long-range connections) theta-band envelopes, but the differences and similarities are not further explained.

Neither do the authors discuss how gamma-band envelopes could become a signifier of long-range connectivity. There must be some form of subcortical or otherwise mediated anatomical pathways. This would probably need to be developed and discussed further in the discussion section.

[page 25]: The analysis of language-dominant vs non-dominant hemisphere is interesting and laudable. Statistically speaking, however, the analysis would deserve proper control for potential other, confounding differences between the two groups of patients.

[page 27]: By and large, the paper appears to argue that the frequency band under consideration is not crucial. This again would also be a somewhat surprising result and would deserve much better documentation/justification than the few sentences about alpha, beta and theta currently provide. 

There is a large body of work, also in this journal (e.g. in recent years Keitel et al., Kluger et al, Alavash et al, in other journals eg Hipp et al, Nat Neurosci) documenting frequency-specific "fingerprints" of different cortical areas. Also in functional networks derived from fMRI data, (much slower) frequency-specific patterns have been exposed. How do the authors reconcile their non-specificity for their DME networks with these findings? (See also my concern about gamma-band envelopes for estimating long-range connections more generally.)

[page 32]: The analyses targeting specific hierarchical relations, e.g. within STS, exposed another problem with the general approach of the present manuscript: The authors do not demonstrate that the found network solutions (ie., clusterings in embedding space) are robust across cross-validated subsets of the data. To the best of my reading, there are no such robustness analyses being reported?

MINOR:

[page 12] The introduction should do more to add justification and specificity: Why the diffusion map embedding (vs many other dimensionality-reduction and graphing algorithms)? 

For a paper striving to provide a definitive answer, the methodological choices (for which there were plenty here) are obviously inseparable from the conclusions to be drawn. As a consequence, major choices like the algorithm become a key feature not a side note and should be justified and interpreted accordingly.

[page 13]: The sentence in line 133f. needs a citation. ("… auditory processing").

Reviewer #3: Review of PBIOLOGY-D-22-02638R1, "Functional geometry of auditory cortical resting state networks derived from intracranial electrophysiology"

In this article, the authors perform a series of analyses to determine the organization of auditory and auditory-adjacent areas in the brain based on their patterns of resting state functional connectivity obtained via intracranial EEG. The overall dataset represents a substantial number of subjects, recording sites, and time points when amassed together. The analyses are reported in a coherent theoretical and empirical framework. Most importantly, the results provide important new insight into the functional organization of the auditory system and, in particular, its integration with the rest of the brain. The is particularly interesting and important when considering the role of audition in the human faculty of language. I note from the manuscript number that this paper is a revision, but want to acknowledge that I am a new reviewer seeing this manuscript for the first time.

This report has numerous strengths. The sample size and density is one. The clear prose and beautiful visualizations is another. In addition, I thought the hemispheric asymmetry analyses were particularly interesting and well executed. Finally, a great strength of the paper is the direct comparison between the iEEG data and the rs-fMRI data from the same participants. Overall, the paper is an empirical and methodological tour de force. Broadly, I think it will make an important contribution to the literature in this area, and will inspire future work based on these methods in other domains.

I have a number of only small comments / recommendations regarding the framing of certain sections:

1) In the Introduction and throughout the paper, I think the claims about lateralization of speech and language could be made with some more nuance. While neurological and neuroimaging evidence do point consistently and uniformly to the leftward lateralization (necessary and sufficient) for language, most evidence, both neurological and neuroimaging, seems to suggest a bilateral organization of speech (perception / recognition). This is most clearly explained in the Hickok & Poeppel (2007) model. Some more nuance in the introduction of this idea could help the paper not come across as overly dogmatic or naive about the bilateral organization of speech. 

2) In the Discussion, I think an important caveat to acknowledge is that the patterns of organization are based on resting state data. It remains an open and unexplored question whether and how these networks are also attested during tasks. There is a lot of thought recently in the literature on dynamic network brain (re)organization / reconfiguration during different kinds of tasks and their difficulty. This is not a criticism of the approach, but it is important to note that these networks are based on a snapshot of behavior (rest) rather that the full, complex range of auditory behaviors.

3) Another minor question in the discussion centers on the hub/spoke distinction. The authors do a good job framing their findings regarding the Hubs, but the brain areas identified as Spokes are not discussed. I wonder if they might comment also on why certain regions (principally somatomotor regions, but also primary auditory regions) appear to be "spokes" in their network organization. 

4) Finally, with regard to the discussion, the authors note differences in the sparseness of network organization in left vs. right hemispheres, and that these are not exclusive to language ROIs. I wonder if they might discuss more about non-linguistic (i.e., domain general) differences in hemispheric functional organization. For instance, classically, a great deal has been made about local vs. global processing in the left vs. right hemisphere, both in vision, space, navigation, etc.

All of these points are quite minor and none of them stand in the way of publishing this impressive work. I learned a great deal reading this paper and think it makes a tremendous contribution to the literature. I want to congratulate the authors on their hard work, compelling findings, and beautiful presentation of these data.

---

## [Decision Letter · Decision Letter 2]

2 Jun 2023

Dear Dr Banks,

Thank you for your patience while we considered your revised manuscript "Functional geometry of auditory cortical resting state networks derived from intracranial electrophysiology" for consideration as a Research Article at PLOS Biology. Your revised study has now been evaluated by the PLOS Biology editors, the Academic Editor and two of the original reviewers.

The reviewers agree that the manuscript has been substantially strengthened in this revision. However Reviewer 1 has a number of additional suggestions to further improve the paper, and we think his/her comments should be carefully addressed. In light of the reviews, which you will find at the end of this email, we are pleased to offer you the opportunity to address the remaining points from the reviewers in a revision that we anticipate should not take you very long (although if you need an extension to our deadline, please do let us know). We will then assess your revised manuscript and your response to the reviewers' comments with our Academic Editor aiming to avoid further rounds of peer-review, although might need to consult with the reviewers, depending on the nature of the revisions.

***IMPORTANT: In addition to addressing the reviewer comments, please attend to the following editorial requests:

1) TITLE: We suggest that the title be edited slightly to be a bit more active. If you agree, we propose it be changed to something like: "Diffusion mapped embedding analyses of intracranial electrophysiology data reveals functional geometry of auditory cortical resting state networks"

2) FINANCIAL DISCLOSURES: Please update your financial disclosures statement, in our online system, to describe the role of any sponsors or funders in the study design, data collection and analysis, decision to publish, or preparation of the manuscript. If the funders had no role in any of the above, include this sentence at the end of your statement: "The funders had no role in study design, data collection and analysis, decision to publish, or preparation of the manuscript."

3) ETHICS STATEMENT: Please update the ethics statement, in your methods section, to include the approval numbers for the protocols approved by the University of Iowa Institutional Review Board and the National Institutes of Health. 

4) DATA AVAILABILITY: Thank you for depositing data used to generate your figures on zenodo. I noticed in your data availability statement that you say the complete dataset is only available with a formal data sharing agreement. Please note that PLOS requires that authors comply with field-specific standards for preparation, recording, and deposition of data when applicable - and so we request that you please upload the complete dataset to a publicly available repository, as Reviewer 2 has suggested.

**IMPORTANT - SUBMITTING YOUR REVISION**

*Resubmission Checklist*

*Published Peer Review*

Sincerely,

Luke

Lucas Smith, Ph.D.

Senior Editor

PLOS Biology

lsmith@plos.org

REVIEWS:

Reviewer #1: The authors made substantial changes to the manuscript. While they strengthen the quality of the study, there is still some concerns that must be taken into account. Please find here some suggestions.

(1) Pertinence of resting state data.

Based on the adding made since the last version of the manuscript, some readers may think that event related paradigms have only cons when it comes to estimate functional connectivity. Although, that are many valuable publications that investigated functional connectivity with active protocols. The sections focusing on the pros of resting states data should be revisited accordingly. To defend their point of view, the authors could explain in the manuscript why they did not analyzed the event-related data to compare them with the resting state result.

(2) Functional connectivity metric.

The use of band-pass filter after estimating the power of each frequency band (line 1062) focuses the signal on the slow fluctuation of each band. This smoothing may have some impact on the present results (e.g., link with cross frequency coupling) or may explain why the results from different frequency bands present some similarities.

As the authors performed the same series of analysis with wPLI, it is worth reporting them, at least in the supplements. 

(4) Threshold.

Some thresholds applied to connectivity matrices are subjective (lines 1066-1073 and lines 1092-1102). It would be critical to provide an objective estimate of the influence of these thresholds.

(6) As discussed thoughtfully in a recent review co-written by more than fifteen iEEG research centers (Mercier et al., 2022), there is no consensus on a given method to perform statistical analysis in the field of iEEG. For instance in the present study, the amount of data differ between participants both in term of data length and in term of contribution to the ROI/parcel (e.g., some result might be driven by a handful of participants that are more represented than the other for a given ROI). Also, it is recommended to provide some estimate of the representativeness or the result or at least to be explicit when 'random-effect' vs. 'fixed-effect' analysis was conducted. 

(7) Most of the study is data driven except the original parcellation defining the ROI. A full data driven approach would have been appreciated. Could the authors discuss the potential influence of the first pooling step?

Minors:

- Blurb: The last sentence could briefly summarize the results rather than suggesting some future applications.

- In the introduction, the sentence related to sensitivity and specificity could be completed with the response made to the previous round of review.

- Can the authors clarify the difference between the terms 'polysensory' and 'transmodal'?

- Line 167: Please provide the metric used to estimate the variance (MAD would be more straightforward to understand than stddev).

- Line 176: This brain parcellation seems to not be described in the materials and methods.

- Lines 186 and 191: The word 'diffusion' might be confusing for the non-specialist as it is mostly used in reference to DTI. Please clarify briefly.

- Lines 192-193: Please add few words to explain how the SNR was computed.

- Figure 1: Maybe the electrodes outside the brain are not needed are they were not analyzed. For the depth (sEEG) electrodes would it be possible to use a depiction that allow their identification? (e.g., transparency or an outline).

- Figure 3: Which estimate of variance was computed? (i.e., stddev, var or another metric).

- Lines 389-391: Please add a sentence to explain how a specific threshold can be chosen depending on the hypothesis testing.

- Figure 4: Panel c is not present in the figure, only in the captions.

- Line 453: The term 'information' flow is misleading as not information metric was used.

- Lines 488, 502 and 511: Only Beta band is mentioned while other frequency bands were analyzed. 

- Line 547: Please provide the number of electrodes used here.

- Line 560: What was the rationale to use the SY parcellation rather than the parcellation used in Figure 1?

- Figure 8: The scaling could be the same for the x axis and the y axis to show that the correlation is above the diagonal.

- Line 765: Why the 'high mean connectivity' was deleted?

- Line 946: Due to the non-linear mapping between an average template and an individual brain, a linear coregistration is not recommended (see Mercier et al., 2022). Can the authors double check this processing step?

- Line 977: The classification of white/gray matter electrodes could be informed by brain segmentation provided by freesurfer. It is a pity that this processing step was not automatized.

- Line 1021: Can the authors provide information about the spatial filter applied to 're-reference' the data?

- Line 1046: This sentence is repeating the one several lines above.

- Lines 1205-1224: Did the target sounds were included in the analysis?

Reviewer #2: Thanks to the authors for such a compelling rebuttal and according revision.

I can follow the authors on a vast majority of points raised, and I consider the revised ms. sufficiently transparent and clear for readers to make up their own mind on a few remaining minor issues of contention.

I look forward to these data being made publically available under a prominent open-source umbrella such as OSF.

Kind regards,

Jonas Obleser

---

## [Editor Report · Decision Letter 3]

7 Jul 2023

Dear Dr Banks,

Thank you for the submission of your revised Research Article "Functional geometry of auditory cortical resting state networks derived from intracranial electrophysiology" for publication in PLOS Biology. Your manuscript has been assessed by the PLOS Biology editorial team and the Academic Editor, and we are satisfied by the changes made in response to the last reviewer comments and our previous editorial requests. Therefore, on behalf of my colleagues and the Academic Editor, Manuel S. Malmierca, I am pleased to say that we can in principle accept your manuscript for publication, provided you address any remaining formatting and reporting issues. These will be detailed in an email you should receive within 2-3 business days from our colleagues in the journal operations team; no action is required from you until then. Please note that we will not be able to formally accept your manuscript and schedule it for publication until you have completed any requested changes.

**IMPORTANT: As you address any formatting and reporting requests to come, we also ask that you also update the ethics statement in your methods section, to include the approval numbers for the protocols approved by the University of Iowa Institutional Review Board and the National Institutes of Health.

PRESS

Sincerely, 

Lucas Smith, Ph.D. 

Senior Editor

PLOS Biology

lsmith@plos.org